# Comparing driving behavior of humans and autonomous driving in a professional racing simulator

**Adrian Remonda**[1]*, **Eduardo Veas**[1,2], **Granit Luzhnica**[2]

**1** Know-Center, Graz, Styria, Austria, **2** Graz University of Technology, Graz, Styria, Austria

* aremonda@know-center.at

## Abstract

Motorsports have become an excellent playground for testing the limits of technology, machines, and human drivers. This paper presents a study that used a professional racing simulator to compare the behavior of human and autonomous drivers under an aggressive driving scenario. A professional simulator offers a close-to-real emulation of underlying physics and vehicle dynamics, as well as a wealth of clean telemetry data. In the first study, the participants' task was to achieve the fastest lap while keeping the car on the track. We grouped the resulting laps according to the performance (lap-time), defining driving behaviors at various performance levels. An extensive analysis of vehicle control features obtained from telemetry data was performed with the goal of predicting the driving performance and informing an autonomous system. In the second part of the study, a state-of-the-art reinforcement learning (RL) algorithm was trained to control the brake, throttle and steering of the simulated racing car. We investigated how the features used to predict driving performance in humans can be used in autonomous driving. Our study investigates human driving patterns with the goal of finding traces that could improve the performance of RL approaches. Conversely, they can also be applied to training (professional) drivers to improve their racing line.

## Introduction

Motorsports have become a professional competition with multiple challenges at technological, engineering and high performance sports psychology levels to name a few. At 300km/h, a car moves 5km per minute and 83 meters per second, and errors of a fraction of a second may well be fatal. Racing drivers have to perform numerous motor and cognitive tasks simultaneously. Focusing on the performance of their vehicle, they shift gears, move foot pedals and steer the vehicle via precise movements while tracking competitors at centimeter distances. The difference between doing a best lap time and an average lap lies often in small variations in braking point or not accelerating aggressively enough. While field engineers in motorsports possess much of this knowledge, formalizing it into concrete methods (e.g., training an autonomous agent for racing) makes it evident that there exists no concrete empirical evidence of

**Data Availability Statement:** All the data from the user study are available as follows: DOI: 10.5281/zenodo.4407140 Target URL: https://doi.org/10.5281/zenodo.4407140 Repository:

https://github.com/dasGringuen/ComparingDriving
BehaviorHumansAutonomousDriving/tree/v1.

**Funding:** The funding for this work was awarded to Adrian Remonda. This research was partially funded by AVL GmbH (www.avl.com) and Know-Center GmbH (www.know-center.at). Know-Center is funded within the Austrian COMET Program - Competence Centers for Excellent Technologies - under the auspices of the Austrian Federal Ministry of Transport, Innovation and Technology, the Austrian Federal Ministry of Economy, Family and Youth and by the State of Styria. COMET is managed by the Austrian Research Promotion Agency FFG (https://www.ffg.at/en). The sponsors and funders played no role in the study design, data collection and analysis, decision to publish, or preparation of the manuscript. The funder provided support in the form of salaries for authors Adrian Remonda, but did not have any additional role in the study design, data collection and analysis, decision to publish, or preparation of the manuscript. The specific roles of these authors are articulated as follows. Each named author (Adrian Remonda, Eduardo Veas and Granit Luzhnica) has substantially contributed to conducting the underlying research and drafting this manuscript. In detail, the three authors contributed to the research methodology and experimental plan. Adrian Remonda implemented the algorithms and executed the study. The three authors contributed to data analysis and reporting. Additionally, the named authors have no conflict of interest, financial or otherwise.

**Competing interests:** The commercial affiliation with AVL GmbH and Know-Center GmbH does not alter our adherence to PLOS ONE policies on sharing data and materials.

how to differentiate top performance driving behaviors. This paper presents a data-driven approach, collecting data from human drivers, deriving features and developing a predictor model to inform the evaluation of a reinforcement learning (RL) system.

Much literature is devoted to the problem of autonomous driving. However, most of the work focuses on passenger cars. The scope of our study is aggressive driving (or racing). In passenger cars the priority is efficiency and comfort, whereas in racing, the ultimate goal is to drive as fast as possible by bringing the car to its physical limits [1–3]. In common practice, this problem is addressed by applying methods from the field of control theory, which utilize heuristics and require domain knowledge to tune the model's parameters manually [4]. Since each racing track has its peculiar challenges, these methods often require a different set of heuristics and parameters for each new situation, which effectively limits their generalization and scalability.

The extensive focus on self-driving passenger cars is fully justified due to the importance of the undertaking and the application scale. However, self-racing is also an interesting and critical problem for several reasons. First, professional racing drivers are a very expensive and inaccessible workforce, whereas reliable self-driving is relatively cheap. Automating racing would allow racing car manufactures and racing teams to test different cars and configurations and identify problems without the need for human drivers. Such tests could potentially be first applied for iterations with self-driving and later involve human driver testing, which could save a lot of costs and human efforts. Second, since the self-racing capabilities are already being tested in competitions, such as Roborace (https://roborace.com/), it makes the self-racing a practical problem that happens to be a very complicated one. Additionally, self-driving passenger cars typically deal with a different driving style compared to racing cars. Different factors affect the complexity and safety of each driving style. For instance, self-racing cars are optimised to drive in conditions where the car is pushed to its limits in terms of driving speed. On their hand, passenger cars are optimised to safely drive while handling passengers, other road objects, traffic lights, different road structures, etc. However, passenger self-driving cars encounter less high-speed situations outside of highways where driving is performed in relatively straight lines. Thus, the models trained for such purpose may not have extensive samples of situations in which the car is pushed to its high speed-limits outside of highways. When confronted with such situations or environments (e.g., accidents or unexpected external events), self-driving passenger cars may lack the necessary experience to react appropriately. Thus, if properly utilized, the models developed for self-racing could benefit the self-driving passenger cars in challenging situations. Last but not least, human drivers could benefit from self-racing agents during the training and preparation process. Not only self-driving agents are inexpensive to train extensively, but they may also introduce strategies (where and when to brake, turn, or throttle) that were not obvious for human drivers and trainers. Thus, strategies derived from very good performance results of a self-driving agent, could be studied by trainers and partially applied by human drivers to increase the performance.

Reinforcement learning aims at training an agent to learn to interact with an environment such as to maximize some notion of long-term reward. Combining RL with deep learning, problems with high-dimensional state spaces can be solved [5, 6]. With algorithms like deep deterministic policy gradient (DDPG), deep RL can be extended to allow for solving continuous action space optimization problems [7]. This is an essential prerequisite for our use case as the racing car expects inherently continuous control (i.e. steering, brake, and throttle) [5, 6]. In this work we apply deep RL directly to drive an autonomous racing car in a professional simulator directly from the car's telemetry data. In recent years RL has become a powerful tool to address high dimensional sequential optimization problems. This type of algorithms excel in performance but have the limitation that a reward function is needed. A reward function is

the objective that the RL algorithm seeks to maximize. The definition of this objective is not always a trivial and a poorly shaped reward function can lead to sub optimal or even bad results (https://openai.com/blog/faulty-reward-functions/). In this work we attempt to address this problem by analyzing laps recorded from humans using the same simulator where we train the autonomous driver. We attempt to identify behaviours that lead to slower and faster lap times and to bring this information to the reinforcement learning objective.

## Contributions

Since our goal was to gather information on how humans drive and bring this to algorithms, we identified the following research questions: a) What control behaviors lead to a better performance in terms of lap time? b) What can be learned from humans that could eventually make our autonomous driver achieve higher performance in less training time? c) Can an autonomous driver be trained using end-to-end reinforcement learning to perform as the highest performance human? What level of performance can be achieved?

To achieve our goal of transferring information learnt from human drivers to RL, we adopted a data-driven approach. This work provides the following contributions:

- A systematic approach to analyzing drivers' behavior in racing.

- An RL driver model that outperformed human drivers in a professional simulator.

- A lap time prediction model highlighting the channels that have significant impact on the final performance. This model can identify channels that are more difficult to learn for humans or AIs.

- A comparison between humans drivers (professionals and amateurs) and an autonomous driver based on the vehicle's control features that can be used to inform the design of the autonomous driver as well as the decisions of human drivers.

- An estimation of the time/laps that the autonomous driver requires to advance through the various performance levels.

The paper is divided in two parts: first, we report a study with users to collect and analyse data. Second, we train reinforcement learning autonomous driver. Part I commenced with the collection of user data. The experiment design involved participants (professionals, amateurs, and beginners) driving in a professional racing simulator. Subsequently, the gathered data was analyzed. Using unsupervised learning techniques, we clustered laps in groups in terms of lap time (from bad to excellent) and spotted patterns that make one lap faster than another with the intention of incorporating this information into the algorithms. In Part II, we trained an RL autonomous driver. RL allowed to minimize the handcrafted design effort since it can learn how to drive just by interacting with the environment.

## Related work

While motorsport is a high paced discipline and as old as computer science, written reports of related studies are scarce [8–10]. Much of the experience in training drivers have focused on sports psychology. In contrast, autonomous racing is relatively new. The emphasis has been placed on finding optimal racing lines and developing process controllers. There are some reports of using RL, albeit not in professional simulators. Additionally, to date there have been no studies comparing human and self-driving agents. Since this study spans over several aforementioned fields, the related work cited below has been grouped according to domain.

## Motorsports

Spackman [11] documented his experience gathered from years of couching Formula 1 drivers and described methods of training them in order to boost their performance. He differentiated the driver's brain as hardware (wetwire) and software. The former is determined by the person's genes and affect reaction times and visual acuity. The latter involves the mental programs that the driver uses to control the car.

Moreover, Spackman subdivided the "software" in two parts: i) a model of the environment, which involves how well the driver can perceive the surroundings and the state of the car (e.g., speed) through his sensors (e.g., vision and the vestibular system); and ii) the ability to plan ahead: once the driver has a model of the environment, he defines a plan of actions in order to reach the objective. The faster and the more accurate the plan is, the better the driver's performance will be. This was supported by Witt et al. [12], Hohlefeld [13], Jeannerod [14] and Parsons [15]. Jeannerod defines motor-simulation (mental programs) as a representation of the future in which the driver imagines a set of actions and their outcome without actually executing them. Parsons shows a solid evidence that humans rely on motor simulation in order to plan actions ahead. Motor simulation ultimately enables drivers to rapidly evaluate actions without actually executing them and allows drivers to explore for optimal actions without taking risks.

Spackman concluded that both hardware and software are important: "no training no winning" and "no genes also no winning". This view contradicts the original work of Ericsson et al. [16], who defined "deliberate practice" as an engagement in highly structured activities that are specifically designed to improve performance in a domain, provide immediate feedback, require a high level of concentration and are not inherently enjoyable. Furthermore, they proposed that expert performance reflects a long period of deliberate practice rather than an innate ability or "talent" [17]. However, more recent work by Hambrick provides new evidence against this view and is in alignment with Spackman, who suggested that some people will never acquire expert performance in some domains, regardless of the extent of deliberate practice they accumulate. His main hypothesis to explain this is that when people are given an accurate assessment of their abilities and of the likelihood of achieving certain goals given those abilities, they may gravitate towards domains in which they have a realistic chance of acquiring expert performance through deliberate practice.

In general, the software part defined by Spackman can be viewed from the ecological psychology perspective of Gibson [18] and could be improved following the process of attunement, calibration and re-calibration [19]. Spackman stated that some drivers can improve their "software" by calibrating or correcting bias and errors in the way they perceive (e.g., optical illusions due to distorted uncalibrated driver sensors). Several studies support the idea of calibration of perceptual-motor skill (mapping between perception and action). Witt et al. [12] concluded that actions and imagined actions directly influence perception. Brand [19] indicated that correcting a bias in the way the drivers perceive the environment requires a re-calibration of the map perception and action. Van Andel [20] compiled a systematic review of a big body of articles on calibration in order to understand in which situations calibration is necessary and when it is more efficient.

Spackman also reported that the drivers' ability to plan can be improved by suggesting them a better racing line. A better racing line and car controls can be a result of a computer simulation and a near-optimal driver model, as the one proposed in this work.

Van Leeuwen et al. [21] were closest to understanding racing drivers. They studied the perceptual and cognitive skills of racing drivers by comparing two groups: racing and non-racing drivers in two experiments. First, they analyzed the reaction time and visual-motor

performance to measure the basic cognitive skill and visual-motor performance. Second, they carried out an experiment on a racing simulator to analyze the drivers' lap times, input controls and racing lines. The results of the first part didn't show any statistical difference between the two groups, indicating that the reaction time is not crucial for performance. However, in the second part racing drivers had better lap times, higher steering activity and a better racing line than non-racing drivers. While we adopted a similar study procedure to collect data from human participants, we extended the work of Van Leeuwen et al. and used insights originating from A.I. methods rather than relying on the analysis from psychological and physiological perspective applied in the above-mentioned works.

## Humans vs machines

Although methods of comparing the performance of humans and machines do exist, there are scarce. Funke et al. [22] completed an extensive analysis of problems that can arise while comparing humans and machines in the context of perception. They concluded that to compare humans and machines accurately, the experimental setting must be such that they alleviate innate differences. They also observed that special care must be taken when making the comparison to prevent any systematic bias by humans. Barrett et al. [23] compared algorithms and humans with regard to visual reasoning tasks. Chollet et al. [24] proposed new methods of benchmarking artificial intelligence algorithms and suggested that the algorithms should focus on how effectively they can deal with different types of tasks rather than each individual task.

Observing and comparing humans and machines in parallel has proven to be beneficial in many situations. For instance, the work of Cichy et al. [25] suggests that deep learning models can be used as scientific models to provide useful predictions and explanations in cognitive science. They also showed that once a good AI model is learned, it can be then explored to create novel ideas and hypotheses.

Generally, benchmarking algorithms with humans as a baseline is a long tradition in AI. It provides a good starting point to address problems. Silver et al. [26] started comparing their algorithm, Alpha Go, first with existing game engines and later with humans amateurs and professionals. Finally, the algorithm was able to beat the best Go player in the world. An exciting outcome of their work is that once the algorithms were better than humans, humans used their output to advance human players' strategy and improve their performance. An analogous scenario but in a different domain motivates the work described in this paper. Mnih et al. [5] also used humans as a baseline for their game-playing algorithms. Spielberg et al. [27] accomplished a similar comparison for their self-driving car performance.

Another common method of using humans to improve algorithms is to imitate human or animal cognition and behaviors. Neural networks of McCulloch et al. [28], the perceptron algorithm of Rosenblatt et al. [29] and the backpropagation algorithm [30] are the fundamental blocks that lead to the deep learning revolution. They were inspired by how the human brain works. By analyzing neuroscience studies, Schaul et al. [31] discovered an experience replay in the hippocampus of rodents, which suggests that sequences of prior experience are replayed during resting or sleep, consolidating the learning. Subsequent replicating of this behavior in reinforcement learning algorithms improved their performance significantly. Schoettle et al. [32] compared human drivers and highly automated cars in urban environments focusing on various aspects of driving, including reaction time, planning, perception sensing capabilities and reasoning. They found out that Machines are generally well suited to perform driving tasks, especially with regard to reaction time, power output and control, consistency, and multi-channel information processing. However, human drivers were still better in terms of reasoning, perception and sensing when driving. Finally [33], applied human driving skills to

improve self-driving car algorithms. They recorded humans driving and identified controller parameters to match the human driving.

## Autonomous driving

In contrast to autonomous racing vehicles whose goal is to drive as fast as possible, autonomous self-driving cars must navigate autonomously in a complex environment without human intervention and require a high level of safety. In autonomous self-driving cars, a lot of attention has been devoted to image processing in order to analyze information based on images from cameras and sensors, e.g., Lidar or Radar [34]. The objective of Koutník et al. [35] was to establish whether a car can be autonomously driven by using images. To that end, they performed a prepossessing step to reduce the state space by representing it in the frequency domain, converting the images into a set of coefficients that were subsequently transformed into weight matrices via an inverse Fourier-type transform. In contrast, Koutník et al. [35] and Jaritz et al. [36] used discrete action space. Our preliminary studies (not included in this paper due to the scope) showed that although in a one-dimensional action space (steering wheel) a discrete action space algorithm such as DQN [5] may learn a policy that allow to drive on the track, but the behavior achieved is far from optimal. Thus, the above preliminary experiments showed that it is unfeasible to drive to the limits by discretizing the throttle and steering. In comparison, human gamers of racing games/sims consider a dedicated steering wheel a worthwhile investment. In professional simulators, such as those used to train F1 pilots, it is a must. In [37–39], the authors attempted to drive end-to-end just by learning from a human demonstration, without manual decomposition into a road or lane marking detection, semantic abstraction, path planning, and control [40]. Tackled the generalization issues in traditional imitation learning for self-driving cars. They used demonstrations based on a conventional proportional-integral-derivative controller (PID controller) that had access to the position of the car with respect to the left and right lanes. Many works from the field of robotics address self-driving cars' challenges by assigning them into categories: localization, scene understanding, path planner and path following. Localization is typically split into two parts: (i) global using GPS; and (ii) local using GPS, cameras, LIDAR, IMUs [33]. [41, 42] were among the earliest works that combined mapping with control engineering. [33] used state estimation and probabilistic models that led to winning the self-driving cars DARPA challenge in 2006.

## Simulators

Typically, simulators differ with regard to the type of input they use and the target application. Below we summarized information about the existing simulators targeting passenger driving and racing.

**Passenger cars.** There are many options when it comes to simulators for self-driving passenger cars [43]. CARLA [44] from Intel is an urban simulator that supports camera and Lidar streams, with depth and semantic segmentation and location information. SUMO [45] is an urban macro-scale modelling of traffic. GAZEBO (ROS) [46] is a 3D dynamic multi-robot physics simulator that supports path planning and vehicle control in 2D and 3D maps. DeepDrive [47] is a driving simulator based on the Unreal Engine. It supports a multi-camera stream with depth using high-performance shared memory. Flow [48] is a multi-agent traffic control simulator built on top of SUMO. Finally, Highway-env [49] is an OpenAI gym-based environment simulator for driving on highways.

**Racing.** TORCS [50] is a very customizable open source 3D racing simulator which supports camera streams and telemetry data. FastLap [51] is a professional 3D racing simulator

that supports cameras, multiple vehicles, driver-in-the-loop and high-resolution tracks. rFpro [52] is a professional 3D racing and urban simulator which supports cameras, driver-in-the-loop, Lidar, high resolution tracks, urban scenarios and emulation of sensors. VSM [53] is a high-end professional vehicle dynamics simulator employed by some Formula 1 teams. It can be used as the physics engine for both FastLap and rFpro.

## Autonomous racing

Attempts to achieve the fastest lap times possible for autonomous racing cars typically combine control theory, determining and utilizing the optimal racing line and/or optimizing the driver model directly. In this context, Brarghin et al. [2] conceptualized the issues associated with the race driver model as: i) identifying the trajectory that leads to the fastest traversal of the track with a given car; and ii) determining the step by step controls or driver inputs needed to instantiate the plan. Cardamone et al. [54] determined the ideal racing line using genetics algorithms via a line-follower bot to measure the lap time. In this context, the lap time is the optimization objective. This method is limited by the quality of the line-follower bot. [55] presented a deep learning approach to generate cost maps learned from human demonstrations, which is subsequently fed into a model predictive control algorithm (MPC) that runs in real time on a real 1:5 scale autonomous vehicle. MPC works by sampling hundreds of trajectories at each step. Once the first step of the sequence is made, the rest of the sequence is discarded and the process begins again. MPC requires a model of the car's dynamics, which can be realized either by first principles or learnt directly from data. In this case, they produced the dynamics model directly from the car's data, while Metz and Williams [1] developed a series of models (tire, vehicle handling, engine / braking models of non-linear differential equations to control an open-wheel vehicle). MPC models can be very computationally expensive when running in real time and the final performance of the model depends heavily on the quality of the learned dynamics model. Although these works represent the state-of-the-art in the field of control theory and it has several advantages, their drawbacks are that the driving performance is limited by the quality of the human demonstrations and that they require a reward function (cost function). Recently, RL has been successfully applied to various complex control algorithms [5, 26]. Reinforcement learning deals with the problem of learning optimal behaviors for an agent to interact with an environment by trial and error. Lillicrap et al. [7] evaluated their deep deterministic policy gradient (DDPG) algorithm on a racing car problem via an open-source simulator, TORCS. They reported that some replicas learned reasonable policies using low-dimensional data and pixels as input. Remonda et al. [3] performed an extensive analysis and extended RL models for autonomous racing. They made several variants of DDPG compete with each other in order to find out which of the methods was most suitable for the task. Besides the algorithms' benchmark, they also found what input state is better not only for the final performance but also for generalizing the different tracks. Both Lillicrap et al. and Remonda et al. used TORCS because its fast processing time which results in quicker iterations while developing the algorithms and makes the evaluation of algorithms manageable. However, TORCS poses many limitations mainly due to the simplicity of the underlying physics engine. On the other hand, a high-fidelity simulator models each individual part of the car with sophisticated models which are then validated with real life data across different types of cars. Additionally, professional simulators allow recording laps driven by humans in a realistic scenario which can be then used a solid baseline to compare the performance of RL algorithms. Using a professional simulator is an important step before deploying the algorithms on real racing cars. In the present study, we relied on a professional simulator and compared the results to those of humans to establish a solid baseline.

## Part 1: Quantifying human driving performance

Our first study was intended to quantify human driving performance in motorsports, with a concrete goal of establishing what makes a great lap in terms of quantifiable parameters that can be used to assess driving performance (e.g., of a human or an algorithm). To this end, we devised a study collecting telemetry data while driving in a simulator.

### Assumptions

The following assumptions were made for the purposes of this study:

- a number of distinct groups in terms of driving performance can be observed

- there are observable differences in vehicle control variables in different performance groups

- these differences can be quantified at varying granularity levels

- with data at the right level of granularity, a predictor of performance can be trained

The ultimate goal of this study was to define a systematic approach to analyzing the driver's behavior during racing.

### Methodology

To address the research goal, we applied the following approach to data collection in the experimental design. The Barcelona circuit (part of the Formula 1 season) was chosen arbitrarily. The official sector and corner division shown in Fig 1 were used in the analysis. The participants were invited to our simulator, which uses a professional physics and graphics engine. After providing consent to participate, the participant filled a general questionnaire with demographic questions and have the driving telemetry recorded, the participants were briefed as to the choice of track and instructed to drive as fast as possible to minimize the lap time while keeping the car on the track. The car is considered out of the track when all four wheels are outside of the track borders, in which case the lap is invalidated. Thereafter, the participants accomplished a two-lap free practice drive to become familiar with the controls and learn the track. The data collection part was split into five sessions. In each session, a participant was asked to drive four laps plus the out lap as fast as possible. Immediately after driving each session, the participant filled a NASA TLX questionnaire.

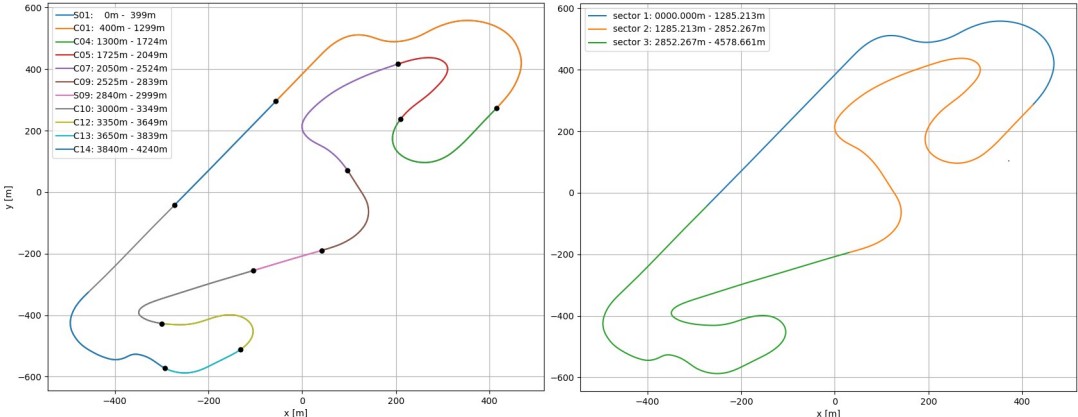

**Fig 1. Barcelona-Catalunya circuit.** The figure shows the track division of the circuit Barcelona-Catalunya. The left part of the Figure shows the corner division. The right one shows Sector I (Blue), Sector II (orange) and Sector III (Green).

## Simulator environment

The software setup consisted of FASTLAP for the visualization and AVL Vehicle Simulation (VSM) for the physics engine. FASTLAP is a real-time driver-in-the-loop driving simulation software for engineering. It is a proprietary software developed by the company Fastlap Simulation Engineering, which offers commercial licenses and a limited academic use. It provides solutions in the fields of motorsports, vehicle dynamics development and research. AVL Vehicle Simulation (VSM) is a professional vehicle simulator provided by the company AVL. For the driver in the loop, the participants drove using a racing steering wheel, racing seat and a projector (Acer Predator Z650). Due to the demanding hardware requirements for running the physics engine and the graphics in real time, the physics engine runs in a dedicated machine and communicates with the graphics engine via UDP. The hardware consisted on an Intel I7 with 32GB RAM for the former and a graphics card Radeon RX Vega with a AMD Ryzen 7 1700X Processor and 32GB RAM for the latter. For the driver in the loop, the participants drove using a racing steering wheel and pedals from Fanatec, a racing seat and a Projector Acer Predator Z650. The gear shift was set to manual and the clutch was automated. No DRS (Drag Reduction System) was allowed. Fuel consumption and tire wear were disabled. The physics engine ran at 2KHz and the telemetry was recorded at 100Hz. The driver model was controlled at 20Hz. Both the autonomous driver and the human participants shared the same track and physics engine, including the car setup.

**Car model.** We used a model of the Dallara F312, which was the official car in 2017 European F3 Open. The Dallara F312 has a Carbon fibre monocoque chassis, a pushrod with twin-damper system and a torsion-bar springs suspension (front), and a pushrod with twin-damper system and a coils springs suspension (rear). The dimensions of the car are: width 1,845 mm (including tires), length 4,351 mm; height 945 mm and wheelbase of 2,800 mm. The engine is a Mercedes-Benz F3 inline-4 engine naturally-aspirated, longitudinally mounted on a mid-engined, rear-wheel drive layout with a 6-speed transmission. The car has a power of 240 hp (179 kW) and a weight of 580 kg, including the driver. The brakes are carbon discs, 6-piston calipers and pads. The tires are: 9" front and 10.5" rear. VSM is capable of modelling all these components. The best lap time in qualification in the 2017 season was 1'38.089, and the worst lap time was 1'40.190. The fastest lap time in the race was 1'40.076. (http://www.euroformulaopen.net/en/, https://en.wikipedia.org/wiki/Dallara_F312).

## Analysis

The vehicle telemetry was recorded in all the sessions. Vehicle telemetry includes numerous channels from which we chose those related to the vehicle control, following the work of van Leeuwen et al. [21]. Performance involves many aspects, such as being fuel-efficient, optimal wear of vehicle's vital parts and optimal tire wear. In this work, the focus was on aggressive driving and driving performance was assessed via the lap time and the mean lap time as well as the sector and segment time. For the purpose of the analysis, the best lap was the fastest lap. The vehicle control includes metrics for steering, brake and throttle (mean, median, max, standard deviation), steering and throttle speed that is indicative of control activity in the corners [56], brake point and brake release relative to the start of the segment and percentage of full throttle. These metrics are complemented with engine RPM, segment entry and exit speeds. In addition, the NASA-TLX questionnaire was used after each session as a subjective measure of self-reported workload in six dimensions: mental demand, physical demand, temporal demand, performance, effort and frustration. The workload was calculated using the NASA TLX-R method of averaging the scores across the dimensions (with inverted performance).

We performed a number of analyses building from the organization of laps at the performance levels, assessing progress of the drivers along these performance levels in different sessions, and finally examined the vehicle control metrics for predicting the performance.

**Drivers' performance analysis**. We analyzed the drivers' performance by session. For each session, a driver was assigned to a performance group according to the average lap times achieved in that section. Under this criterion, a driver could have a few poor runs and be classified into a good group after achieving the best lap time. We assessed differences in the subjectively assessed (NASA TLX) workload per session against the performance the group achieved. Moreover, we observed the evolution of drivers in the performance groups across various sessions.

**Vehicle control features**. The vehicle control features were calculated at several granularity levels: full lap, sectors and segments. Using the official sector division of the track, we coarsely divided the track into eleven segments (nine corners and two straight) and a finer one via moving windows. We observed which vehicle control features differed at the various performance levels.

**Performance prediction**. Finally, we developed a method of predicting performance using vehicle control features. We analyzed which features provided better information for predicting the performance and tested those via a subset of laps from each performance level.

## Participants

Thirteen (13) participants were recruited for the study. Two of them were professional drivers working with professional racing simulators and with experience in international racing (Formula 3). Eleven participants were recruited from engineering and computer science groups at the university campus. Since we had limited access to professional drivers, the sample size of the participants' group was determined by their experience in different modalities of racing. However, this did not affect the results, given that the focus of our study was the quality of laps rather than the qualifications of drivers. The distribution of participants across the modalities of racing was as follows: 2 participants had experience with driving real racing cars, racing simulators, racing games and go-karts; 3 participants had experience with racing simulators, racing games and go-karts; 3 participants had experience with racing games and go-karts; 2 participants had experience only with go-karts; 3 participants had no experience in racing whatsoever. As a result, the sample size was determined by the availability of professional drivers, which was sufficient for racing more than 40 laps altogether. In total, the 13 participants performed 258 laps. Before starting, the study participants completed a consent form and an intake questionnaire consisting of demographic questions. First, general questions were posed about driving passenger cars. 100% had a driving license and drove on average 5.08 (SD = 4.5) hours a week. They had on average 13.3 (SD = 9.19) years of experience driving and they do on average 9165 (SD = 8077) annual kilometers. They had on average 0.76 (SD = 1.16) number of accidents and on average 14.69 (SD = 30) traffic fines. In addition, the participants completed a self assessment of their driving skills on the scale from 0 to 10, with 10 being the best, resulting in an average of 7.23 (SD = 1.96). Second, they were asked questions about driving racing cars. They have on average 0.59 (SD = 1.94) years of experience and participated on average in 5.46 (SD 7.14) go-kart races. They drove on average 0.36 (SD = 0.76) hours a week using a racing simulator and played racing games on average 0.93 (SD = 2.28) hours a week. Finally, the participants provided a self assessment of their driving skills in video racing games and on a simulator, which resulted in 5.42 (SD = 1.98) and 5.46 (SD = 2.14) on average, respectively.

**Table 1. Overall performance statistics.**

| trajectory | lap | s1 | s2 | s3 | S01 | C01 | C04 | C05 | C07 | C09 | S09 | C10 | C12 | C13 | C14 |
|---|---|---|---|---|---|---|---|---|---|---|---|---|---|---|---|
| min | 99.1 | 22.9 | 34.0 | 40.8 | 6.0 | 16.9 | 8.6 | 8.1 | 10.4 | 5.9 | 2.7 | 6.6 | 8.1 | 3.5 | 6.4 |
| median | 117.4 | 26.5 | 40.8 | 47.7 | 6.2 | 20.6 | 10.5 | 9.7 | 12.2 | 7.2 | 3.0 | 10.1 | 9.3 | 5.5 | 13.6 |
| max | 211.3 | 58.1 | 110.0 | 124.7 | 17.9 | 40.7 | 69.8 | 63.0 | 29.8 | 75.7 | 17.0 | 17.0 | 37.0 | 32.9 | 84.6 |
| mean | 121.0 | 27.3 | 42.6 | 50.3 | 6.5 | 21.0 | 10.9 | 10.5 | 12.6 | 7.9 | 3.2 | 10.1 | 9.9 | 5.9 | 15.3 |
| stdev | 18.9 | 3.9 | 9.4 | 10.1 | 1.1 | 3.2 | 4.0 | 4.5 | 1.9 | 4.7 | 1.1 | 1.3 | 2.7 | 1.9 | 7.5 |

The table contains performance statistics (times in seconds) for the complete circuit, for sectors (s1-s3) and segments (S01-C14), including the min. (best time) median, max. (worst time), mean and standard deviation.

## Results

In the study, a total of 13 participants x 5 sessions x 4 laps per session = 260 laps were expected. One of the professional participants completed two fewer laps than expected, reducing the total number of laps to 258. Table 1 summarizes the statistics per lap sector and segment. The difference between max and median indicates that there is a number of poorly driven laps (slow laps). In the following analysis, these laps are not removed but rather are considered as such that a measure of slow laps is present as well.

**Performance levels.** Aiming to obtain metrics for performance in racing beyond merely the best lap times, we attempted to classify the laps into the performance levels. To do so, three clustering approaches were tested to group the laps according to the performance using: i) a 1-dimensional lap descriptor laptime; ii) a 3-dimensional lap descriptor consisting of sector times $l = \{s1, s2, s3\}$; iii) an 11-dimensional lap descriptor via segment $l = \{S01, C01, C04, C05, C07, C09, S09, C10, C12, C13, C14\}$. Although all of these approaches rely on the performance measured in time as a base, ii) and iii) may account for variability in the performance at higher granularity levels. In all cases, analysis via the elbow and silhouette methods indicated that the preferred number of clusters is between four and six. The number was confirmed using the Nbclust in the R package, which provides 30 indices to determine the number of clusters in a data set [57]. Solutions with a low number of clusters detected the differences in bad laps but created large clusters for fast and very fast laps. Therefore, we opted for a solution with six clusters using K-Means. Table 2 shows the statistics for the clusters ordered from the *WORST* to *EXCELLENT* lap times. The cluster names refer to the lap-time, i.e., the worst means the worst lap-time and excellent means an excellent lap-time. Fig 2 illustrates these clusters as they occur in each session. The *WORST* cluster occurs in the first sessions, quickly declines until session 3

**Table 2. Laptime statistics for performance levels.**

| Group | # Laps | mean | std | min | %25 | %50 | %75 | max |
|---|---|---|---|---|---|---|---|---|
| Worst | 7 | 193.155 | 11.1506 | 176.738 | 188.543 | 191.017 | 197.99 | 211.266 |
| Bad | 14 | 154.071 | 5.43107 | 146.684 | 150.756 | 153.388 | 155.509 | 168.442 |
| Medium | 36 | 137.371 | 3.33668 | 131.255 | 134.891 | 137.536 | 139.718 | 145.716 |
| Good | 61 | 124.569 | 3.37206 | 119.241 | 121.707 | 124.773 | 127.356 | 130.372 |
| Very good | 73 | 113.372 | 3.14381 | 108.463 | 110.962 | 113.277 | 115.877 | 118.825 |
| Excellent | 67 | 102.844 | 3.17731 | 99.08 | 99.8 | 101.68 | 106.105 | 107.68 |

Laptime statistics for performance levels. The values are in seconds.

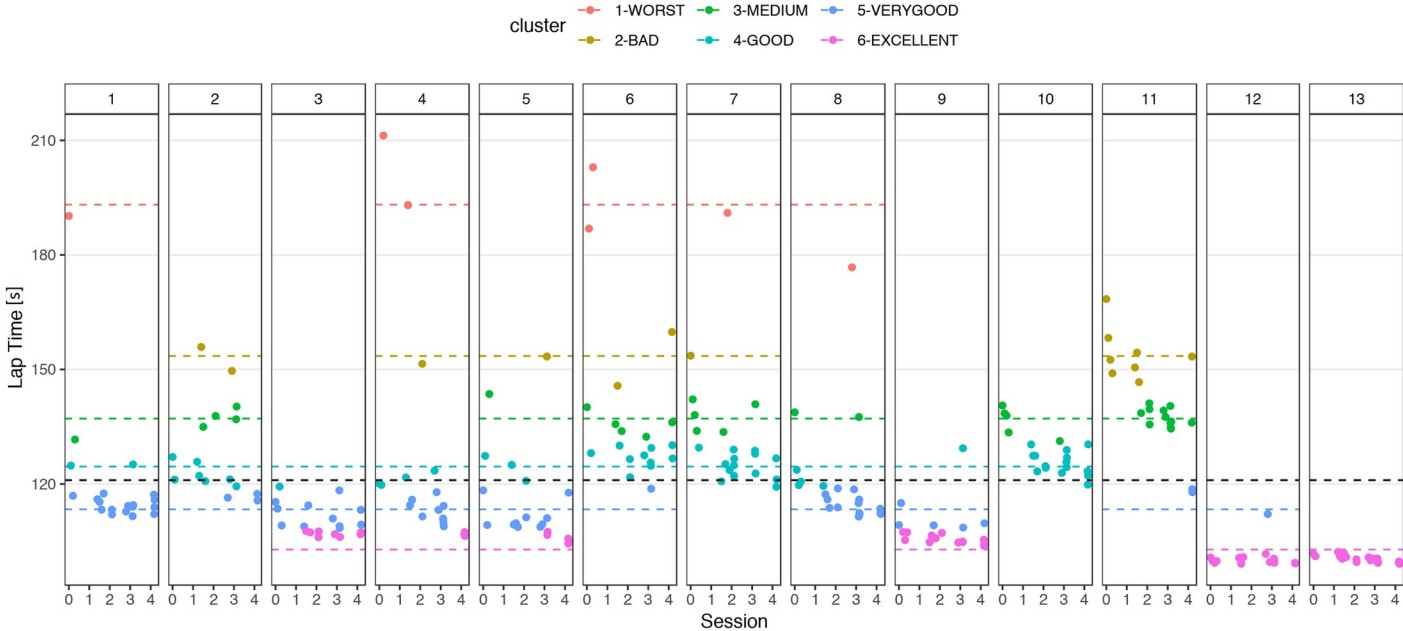

**Fig 2. Lap times by user, session and cluster.** Each dot represents a lap, the position in the user box depends on the session and the color depends on the performance level. The grand mean is plotted in black dashed line and the group means are plotted in dashed lines with group color.

and disappears in sessions 4 and 5. The fastest laps in *VERYGOOD* and EXCELLENT occur in session 4. The cluster with the fastest lap time is relatively stable.

**Driver performance analysis.** The drivers had varying performance throughout the sessions (see Fig 2). In general, however, they showed progressive improvement; with *POOR* lap times in the initial sessions and increasingly faster ones from the third session on. One participant (12) achieved only *EXCELLENT* lap times and two participants (3, 9) mostly *VERYGOOD* or *EXCELLENT* lap times. Other participants had laps spread across three or even four performance groups, with only sporadic *VERYGOOD* and even fewer *EXCELLENT* lap times. In each session, a participant would drive laps at difference performance levels. The performance levels were averaged to obtain a notion of how well the driver did in that session. In the first session (0), two participants had a *BAD* performance, four had a *MEDIUM* one and two in each had *GOOD, VERYGOOD, EXCELLENT* ones. In the second session (1), one participant had a *BAD* performance, three a *MEDIUM* one, two had a *GOOD* one and three had *VERY-GOOD, EXCELLENT* ones. In the third session (2), one participant had *MEDIUM* performance, five had a *GOOD* one and three in each had *VERYGOOD, EXCELLENT* ones. The fourth session (3) had two participants in the *MEDIUM* category, three in the *GOOD* one, six in the *VERYGOOD* one and one in the *EXCELLENT* one. The fifth session (4) had one *MEDIUM*, three *GOOD* and *VERYGOOD* and five *EXCELLENT* performance categories.

After each session, the participants filled a NASA TLX questionnaire to assess the workload. Due to our method for classifying driving performance, it is not possible to ensure balanced groups and the group assignment changes from session to session, as described above. Still, plotting the TLX dimensions per session and cluster (i.e., taking the participants in the session that performed at each performance level in that session) the TLX can be plotted as shown in Fig 3. Interestingly, the participants in the *EXCELLENT* performance level were confident about their performance and reported lower Effort and Frustration, as well as lower Mental, Physical and Temporal demands. Conversely, participants at the *VERYGOOD* performance

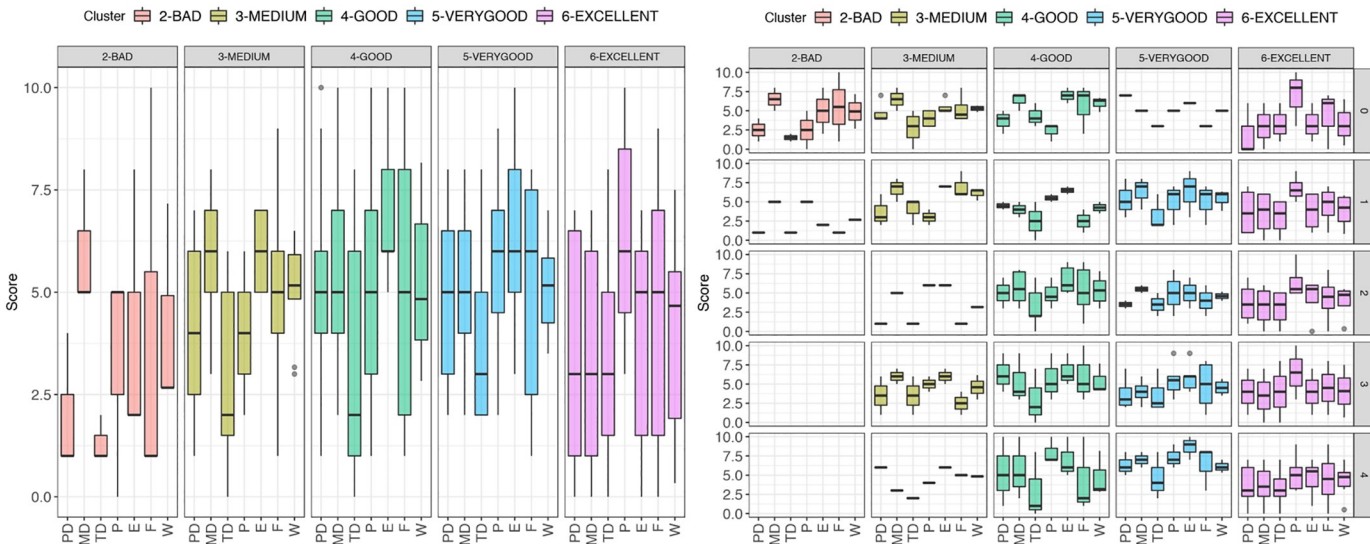

**Fig 3. TLX by performance level.** Users were assigned to a performance level according the average lap assignment. No user was assigned to a level *WORST*. Starting with session 3 no user was assigned to level *BAD*. (Left) Summary TLX, performing at *EXCELLENT* level reports lower Effort than *GOOD* and *VERY GOOD*. (Right) TLX per group and session illustrates apparent difference in reported Mental Demand and Frustration between *EXCELLENT* and *VERY GOOD*.

level reported more effort and frustration, even more so than the participants at worse performance levels, such as *GOOD* and *MEDIUM*. Although the data are too sparse for inferential statistics by session, they can be compared on the general level. Comparisons on Mental Demand, Frustration and perceived Performance revealed only small effects, but perceived Effort had a large effect. A Kruskal Wallis test revealed a significant effect of Performance Level on *TLX Effort* ($\chi^2(4) = 20, p < 0.01$). Post-hoc Dunn tests (Bonferroni adjusted) revealed significant differences between Levels 6 and 4 ($p < .02, z = .56$), and between Levels 6 and 5 ($p < .01, z = .64$). These statistics confirm the above claims. Other tests did not suggest significant differences.

**Vehicle control features.** Fig 4 shows a typical chart used in motorsports to analyze driving behavior. The chart overlaps the best laps from each performance level, illustrating the speed, the steering angle, the brake pedal position, and the throttle as a function of traveled distance in the lap. The chart illustrates how the *EXCELLENT* drive gains speed immediately after each corner, accelerating earlier and more aggressively, with a sharp (almost binary) treatment of the throttle pedal and steady handling of steering. Fig 4 demonstrates several differences in the telemetry of laps at different performance levels. Fig 5 compares all laps in groups *VERYGOOD* and *EXCELLENT*. The latter group released the throttle and brake later and pushed the throttle earlier and sharper.

We intended to capture differences in the vehicle control channels (steering wheel, throttle pedal and brake pedal) using machine learning methods. To do so, we extracted a total of 60 statistical features which were subsequently reduced to 33 by removing highly correlated features. Fig 21 illustrates the break-down analysis by cluster of a sample of these features. Higher performing groups have a higher percentage of full throttle across all sectors and a higher exit speed in sector I but not in sectors II and III. The maximum and mean RPMs tend to be higher for faster laps across all the sectors. The first quartile of the throttle is higher in faster groups regardless of the sectors. In the brake features, the brake maximum and median are higher for laps of the higher performing groups. The brake mean follows the same pattern. The steering wheel features did not show any significant trend (see Appendix).

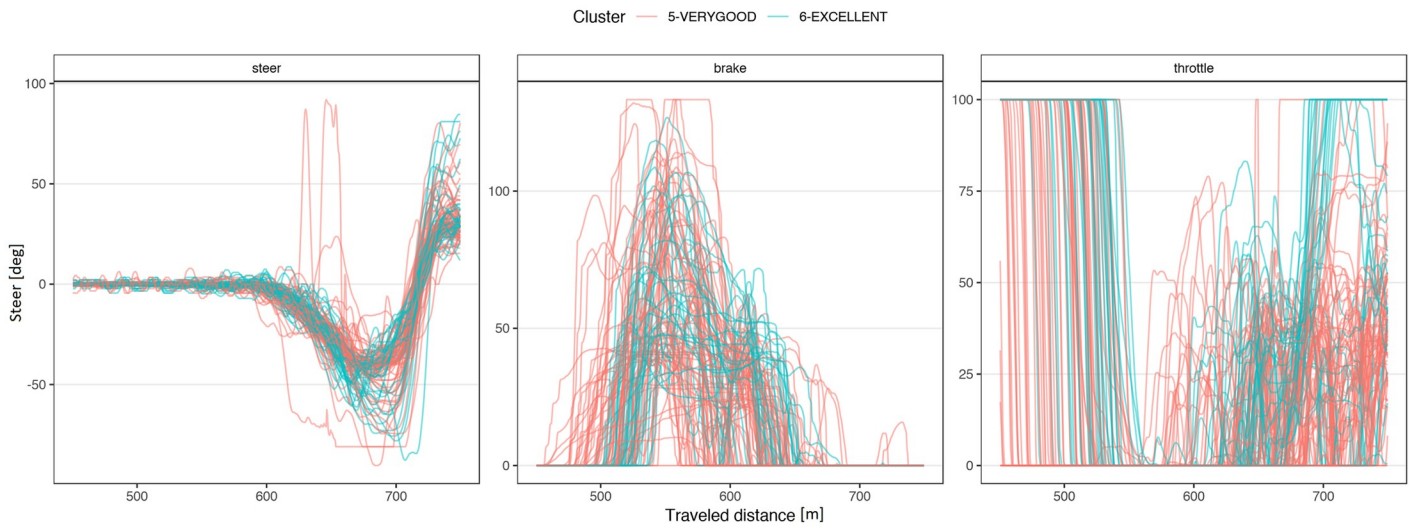

**Fig 4. Best lap times.** Best lap times by performance level.

**Fig 5. Corner 1 comparison.** Telemetry of all laps from VERYGOOD group (red) and EXCELLENT (cyan) group split into steering, brake and throttle.

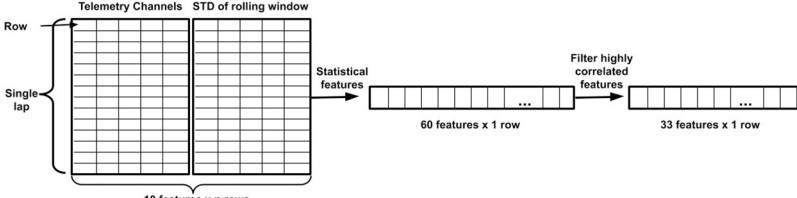

**Fig 6. Feature calculation.** Block diagram of the features calculation process for predicting the lap performance based on the driving behaviour.

## Analyzing driving patterns to predict performance

Considering that all the drivers shared the same car and track, the performance of a driver during a lap (lap-time) is a reflection of the driver's actions took during that lap and the driving patterns (braking, throttling, steering and car speed). Thus, by assessing the driving patterns and the driver's behavior during the lap, the performance during a lap can be predicted, as discussed below in this Section. Predicting performance based on features of the actual telemetry is useful if it can be transferred to autonomous vehicles (see section: "Comparing Driving Behaviors of Humans and RL Agents").

We relied on telemetry data that contained several channels, including the speed of the vehicle. Aiming to analyze driving behavior, we utilized only the parameters that the driver can directly control and their derivations. Since the driver can control the throttle, brake, gear and steering, in our analysis we utilized the following channels: the brake, the throttle, the steer, the engine rpm and the steering change. The engine RPMs describes the engine's rotations per minute and is a function of the gear and the throttle pedal position. The steering change is calculated as the first derivative of the steer variable multiplied by a factor and it is measured in degrees per second.

One important aspect of controlling the car is how much variation or consistency exists within those controls. For instance, braking lightly but often or braking sharply but rarely may not only reduce the speed before a corner, but also have a different impact on lap time. To capture patterns of such variations in controls, we used a sliding window of 200 points (which at a frequency of 100 Hz translates to 2 seconds) for each of the five channels and then computed the standard deviation for such windows. In total, adding the standard deviation of rolling windows results in 10 features for each measurement point during a lap. On top of the 10 features of the entire lap, we calculated statistical features for the entire lap, such as the mean, the standard deviation, the median, 25th quantile, 75th quantile and the maximum. This resulted in 60 features for each lap in total. We performed a feature filtering process to remove features that highly correlate with each other (the absolute value of the Pearson correlation coefficient > .9) by keeping only one of the correlated features. After the filtering process, only 33 features remained for each lap and were utilized to predict the completion time of a lap. The entire feature engineering process is illustrated in Fig 6.

To perform a lap time prediction, we considered several models. The details of such 591 exploration will be omitted due to the scope of the paper, but we decided to use a linear regression model due to its prediction accuracy. To train the model, we split the data into a training set (70%) and testing set (30%). The splitting was balanced by the lap time. The training and testing sets were used to train the model and to evaluate the predicting accuracy, respectively.

Fig 7 illustrates the prediction of lap-time for both training and testing sets and presents the histogram of the prediction error (test set only). The linear regression model trained via the training set using the engineered feature can predict the lap time with an absolute accuracy of

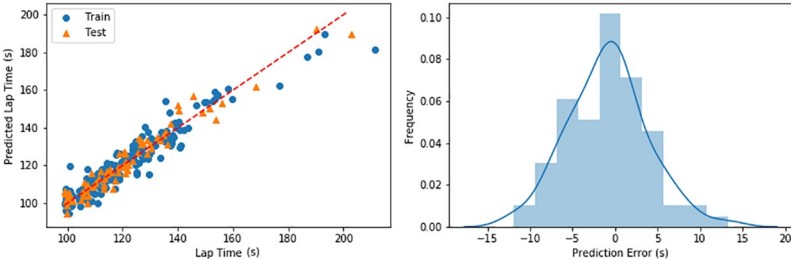

**Fig 7. Lap time prediction.** Linear regression predicting the lap time of human drivers using selected features (left), histogram of prediction error (right) on the test set.

97% or an absolute error of 3.64 seconds (3%). The weights of each feature for the linear regression model presented in Table 4 demonstrate how each feature affects the predicted lap time. Negative weights mean that they contribute to shorter lap time, positively affecting the driving performance and vice versa.

In addition, we explored whether it would be feasible to predict the lap-time by only examining the telemetry data for a single segment rather than the data for the entire lap. For instance, it would be interesting to establish whether some parts of the lap are critical for the performance during the entire lap. The results of this analysis are presented in Table 3, which shows how well the linear regression model predicts both the entire lap time and the segment time based on the features extracted only from that segment. Interestingly, the telemetry in some segments (e.g. S01, C01, C04) can predict the entire lap time quite accurately even before the lap is finished. Perhaps such segments are the ones that make the difference in the driving performance. In addition, we also explored how well such predictions could be made by considering data from the start of the lap until the given segment (see Start to Segment in Table 3).

## High vs poor performance driving patterns

In the previous Section, we discussed the features that are crucial for predicting the performance of a lap. Other important questions are how good human and agent drivers differ from

**Table 3. Absolute error in seconds (mean and std) for predicting the lap time and segment time based on the features extracted form segment(s).**

| Start to Segment | Lap Time | Segment Time | Segment | Lap Time | Segment Time |
|---|---|---|---|---|---|
| **Start-S01** | 4.825 (4.098) | 5.144 (5.658) | **S01** | 4.825 (4.098) | 5.144 (5.658) |
| **Start-C01** | 4.901 (4.564) | 5.068 (5.182) | **C01** | 10.88 (13.673) | 1.126 (0.931) |
| **Start-C04** | 5.585 (6.678) | 4.781 (5.049) | **C04** | 8.80 (13.889) | 1.511 (6.224) |
| **Start-C05** | 5.159 (11.985) | 4.684 (4.496) | **C05** | 7.987 (8.954) | 1.22 (1.242) |
| **Start-C07** | 5.317 (3.938) | 6.070 (6.421) | **C07** | 9.043 (10.532) | 0.777 (0.614) |
| **Start-C09** | 4.332 (12.432) | 4.833 (3.951) | **C09** | 11.588 (14.949) | 1.275 (1.824) |
| **Start-S09** | 4.170 (3.998) | 5.305 (5.266) | **S09** | 43.295 (282.754) | 9.203 (79.863) |
| **Start-C10** | 16.092 (63.156) | 4.407 (11.133) | **C10** | 11.058 (14.672) | 0.612 (0.92) |
| **Start-C12** | 4.241 (11.368) | 4.597 (4.466) | **C12** | 9.636 (12.400) | 0.889 (1.033) |
| **Start-C13** | 4.146 (12.201) | 5.500 (4.426) | **C13** | 11.429 (14.289) | 1.348 (6.959) |
| **Start-C14** | 4.873 (5.253) | 4.403 (3.926) | **C14** | 8.460 (9.622) | 2.121 (2.645) |

Left side (Start-Segment) presents the prediction result based on the features calculated from the start point (start of S01) till the segment, including the segment. For instance Start-C04 includes segments S01, C01 and C04. The rights side presents the prediction using the features extracted only from the given segment.

**Table 4. Feature weights of the linear regression models showing how each feature affected the lap time prediction for human drivers (HD on the left) and RL agents (RL on the right).**

| Feature | Weight (HD) | Positions | Feature | Weight (RF) |
|---|---|---|---|---|
| throttle_mean | -10.77 | ↑\|↑ | brake_median | 10.39 |
| engineRPM_mean | -8.71 | ↑\|↑ | rolling_brake_median | -2.71 |
| brake_mean | -8.36 | ∼\|∼ | brake_mean | 2.41 |
| rolling_steer_mean | -7.12 | ↑\|↑ | brake_max | -2.00 |
| throttle_q25 | 7.06 | ↑\|↓ | throttle_mean | -1.04 |
| steer_std | 5.14 | ↑\|↑ | rolling_brake_q75 | 0.37 |
| rolling_brake_q75 | 5.06 | ↓\|↑ | rolling_throttle_mean | -0.22 |
| throttle_max | 4.80 | ↑\|↑ | rolling_steer_q25 | 0.13 |
| engineRPM_std | 3.87 | ↑\|↑ | rolling_throttle_q25 | 0.11 |
| rolling_throttle_mean | 3.55 | ↓\|↓ | steer_std | 0.10 |
| rolling_throttle_median | -3.40 | ↑\|↑ | steer_speed_median | -0.09 |
| steer_speed_std | 2.85 | ↑\|↑ | steer_q75 | 0.06 |
| engineRPM_max | -2.71 | ↑\|↓ | rolling_steer_mean | 0.06 |
| steer_max | -2.69 | ↑\|↑ | steer_q25 | 0.06 |
| brake_median | -2.40 | ↓\|↓ | throttle_q25 | 0.05 |
| throttle_std | 2.20 | ↑\|↑ | steer_mean | -0.05 |
| rolling_throttle_q25 | -2.05 | ↓\|↑ | rolling_steer_median | -0.04 |
| rolling_steer_q25 | -1.97 | ↓\|↓ | rolling_throttle_median | -0.03 |
| steer_median | -1.92 | ↑\|↓ | throttle_std | 0.02 |
| rolling_engineRPM_mean | 1.89 | ↑\|↑ | rolling_steer_max | 0.02 |
| rolling_brake_median | 1.88 | ↓\|↓ | engineRPM_mean | -0.01 |
| steer_mean | 1.35 | ↓\|↑ | steer_speed_mean | 0.01 |
| steer_speed_q25 | 0.88 | ↑\|↓ | rolling_engineRPM_mean | 0.01 |
| steer_q25 | -0.80 | ↓\|↓ | steer_median | -0.01 |
| rolling_steer_max | 0.34 | ↓\|↑ | rolling_engineRPM_std | 0.00 |
| steer_speed_mean | 0.32 | ↓\|↓ | engineRPM_std | 0.00 |
| brake_max | 0.31 | ↓\|↓ | engineRPM_max | 0.00 |
| rolling_engineRPM_std | 0.30 | ↓\|↑ | steer_speed_max | 0.00 |
| rolling_steer_median | -0.18 | ↓\|↓ | steer_speed_std | 0.00 |
| steer_q75 | -0.11 | ↓\|↑ | steer_speed_q75 | 0.00 |
| steer_speed_max | -0.07 | ↓\|↓ | steer_speed_q25 | 0.00 |
| steer_speed_q75 | -0.02 | ↓\|↓ | steer_max | 0.00 |
| steer_speed_median | 0.00 | ↓\|↓ | throttle_max | 0.00 |

The absolute value of the weight emphasizes how important a feature is for predicting the lap time. The features are sorted based on absolute weight, meaning that the earlier a feature shows on the list, the more it affects the lap time. The field positions show whether the feature (on the left or right) is listed earlier (↑), later (↓) or in the same (∼) position in the Table when comparing human vs RL agents drivers.

The weights belong to two trained models: (i) from human laps and (ii) RL agent laps. Negative weights mean that they contribute towards a lower lap time, positively affecting the driving performance.

bad drivers and how human drivers differ from agent drivers while achieving comparable performance.

Figs 8–10 show sectors I, II and C10 of the Barcelona circuit, respectively. They provide a comparison between human driver laps from groups EXCELLENT and WORST. It is obvious that both groups of drivers perform similarly when driving straight and the entire performance difference is due to their handling of corners. Comparing the driving behaviors with the

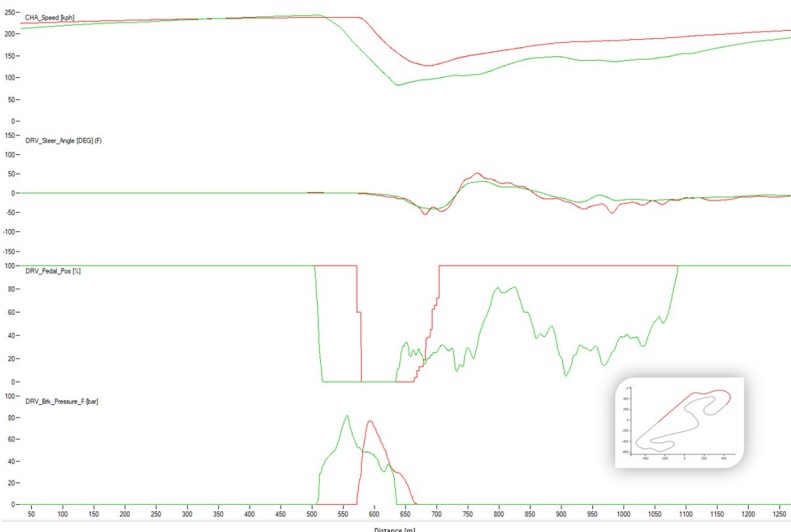

**Fig 8. Comparison between a lap from the human group EXCELLENT (red) and a lap from the human group BAD (green).** The image shows the corner 1 of Barcelona. The X-axis represents distance through the racing line. The Y-axis (from top to bottom): Vehicle speed, steering wheel angle, throttle position, brake position, front left tire saturation, lateral acceleration, longitudinal acceleration.

feature weights of the linear regression presented in Table 4 indicates a clear consistency. The results in Table 4 suggest that laps with low lap-times have features with a high throttle, RPMs and brake mean, a low percentage of 25% percentile on the throttle and a low steering wheel standard deviation. All these patterns are reflected in the Figs 11–13. For instance, Fig 11 shows that a good driver keeps the throttle at the maximum much longer (which results in a

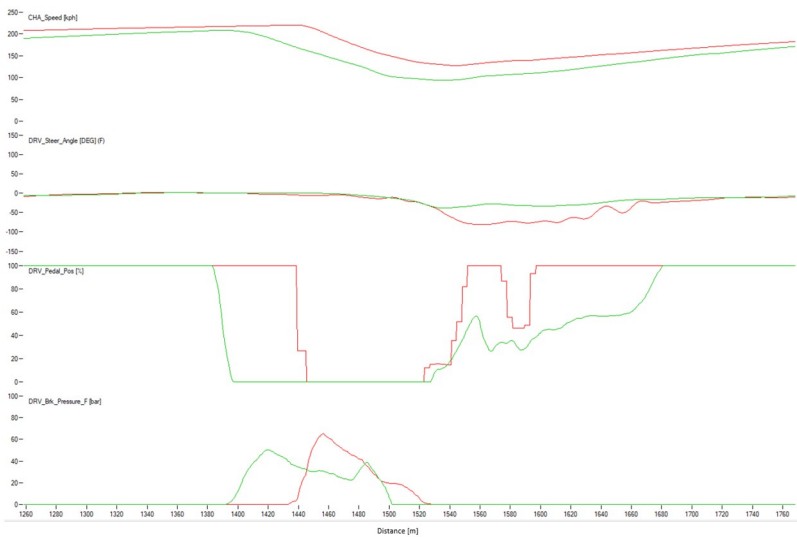

**Fig 9. Comparison between a lap from the human group EXCELLENT (red) and a lap from the human group BAD (green).** The image shows the corner 3 of Barcelona. The X-axis represents distance through the racing line. The Y-axis (from top to bottom): Vehicle speed, steering wheel angle, throttle position, brake position, front left tire saturation, lateral acceleration, longitudinal acceleration.

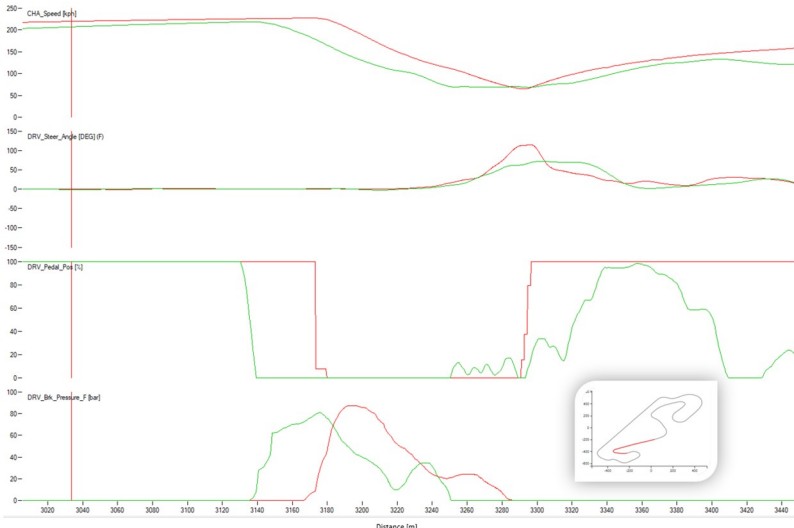

**Fig 10. Comparison between a lap from the human group EXCELLENT (red) and a lap from the human group BAD (green).** The image shows the corner 10 of Barcelona. The X-axis represents distance through the racing line. The Y-axis (from top to bottom): Vehicle speed, steering wheel angle, throttle position, brake position, front left tire saturation, lateral acceleration, longitudinal acceleration.

higher mean value) and brakes less often (which results in a lower mean Value) than a bad driver.

Similarly, comparing RL agent laps (Figs 11–13) shows that agents that maintain a lower median brake, a higher rolling median brake, a lower mean brake and a higher throttle perform better (see Table 4). Interestingly, brakeing has the most impact on the performance of the lap driven by an RL agent. This suggests that the agent needs more time to learn to brake, meaning that this control is the hardest to master. This can be explained via the reward

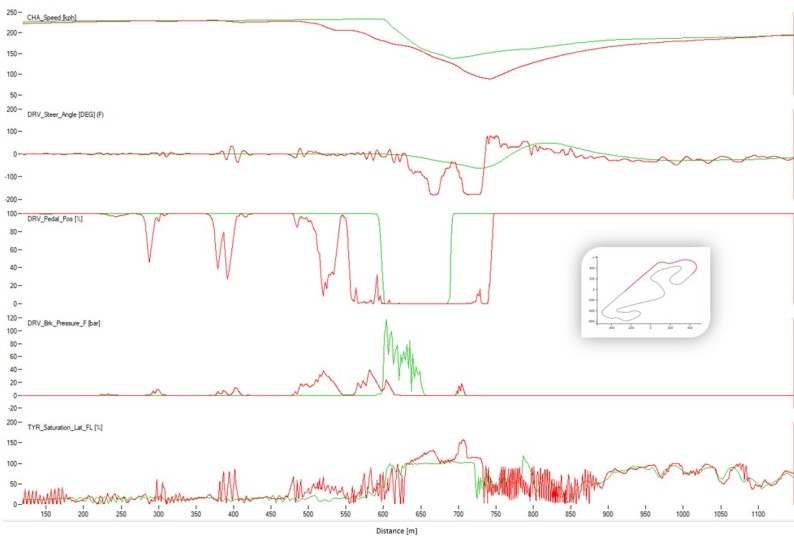

**Fig 11. Comparison between a lap from the AI group EXCELLENT (green) and the AI group BAD (red).** The image shows the corner 1 of Barcelona. The X-axis represents distance through the racing line. The Y-axis (from top to bottom): Vehicle speed, steering wheel angle, throttle position, brake position, front left tire saturation, lateral acceleration, longitudinal acceleration.

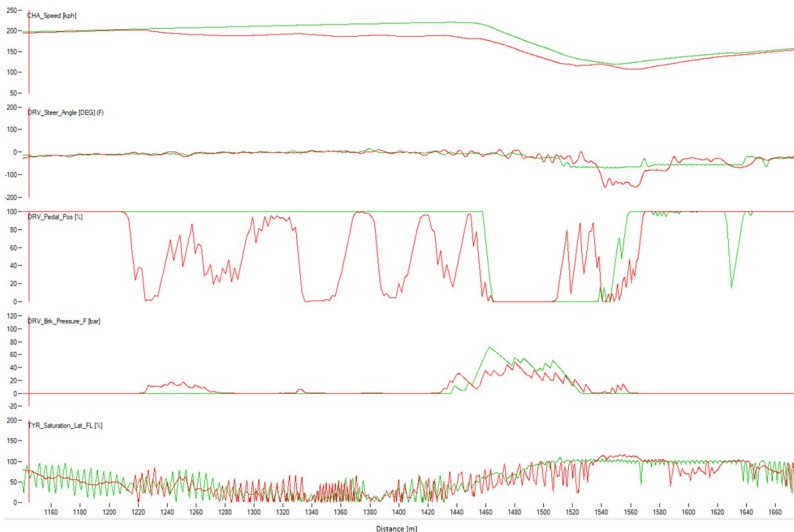

**Fig 12. Comparison between a lap from the AI group EXCELLENT (green) and the AI group BAD (red).** The image shows the corner 3 of Barcelona. The X-axis represents distance through the racing line. The Y-axis (from top to bottom): Vehicle speed, steering wheel angle, throttle position, brake position, front left tire saturation, lateral acceleration, longitudinal acceleration.

function since the RL agent aims to maximize the speed when braking and thus, the model gets a low reward short-term but a high reward long-term. Figs 11 and 13 show that a good RL agent driver brakes less often but more sharply, which results in a lower median of brake and a lower average of brake (since it brakes less often) but in a higher median of rolling standard deviation for the brake (since it brakes more sharply). Additionally, similarly to human drivers,

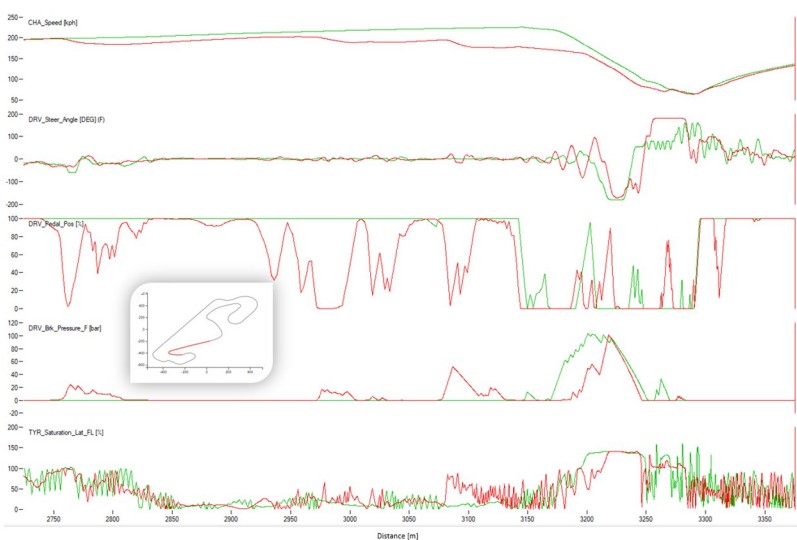

**Fig 13. Comparison between a lap from the AI group EXCELLENT (green) and the AI group BAD (red).** The image shows the corner 10 of Barcelona. The X-axis represents distance through the racing line. The Y-axis (from top to bottom): Vehicle speed, steering wheel angle, throttle position, brake position, front left tire saturation, lateral acceleration, longitudinal acceleration.

a well performing agent driver keeps full throttle longer, which is also evident on the features weights presented in Table 4.

The agreement between the results in Figs 11–13 and the weights of the model in Table 4 further validates our method, clearly confirming that our extracted features can accurately capture driving behaviors of both human and RL agent drivers.

## Autonomous driving

This Section describes the autonomous driver that was trained using RL. It is split in two parts. The first one describes the algorithm and the training procedure. The second one analyzes the results and compares them to the results in the first study.

### Background

RL deals with the problem of learning optimal behaviors for the interaction of an agent with an environment by trial and error. That is, the agent observes some state in the environment and chooses an action, the result of which is another state and a reward obtained from the environment. The agent attempts to learn behaviors in order to maximize the accumulated reward obtained from the environment. Formally, the interaction of the agent with the environment takes place in discrete time steps $t$. At each step, starting from a state $s_t$, the agent executes an action $a_t$ and receives a reward $r_t$ and a new state $s_{t+1}$ from the environment. The return from a state is defined as the sum of discounted future reward, $R_t = \sum_{i=t}^{T} \gamma^{(i-t)} r_i$, where $\gamma \in (0, 1]$ is a discount factor and $T$ is a terminal time step after which the process restarts. The objective of RL is to learn a policy $\pi$, mapping states to actions that maximize the return from the start distribution. Note that the state and action are characterized by the number of dimensions and the type of values each dimension may adopt. If dimensions of the state assume discrete values, it is referred as a discrete state space or continuous state space otherwise. Similarly, if the dimensions of actions assume discrete values, we refer to a discrete action space or continuous action space. The type of state and action space has important implications to the sampling strategy used for example, to choose actions. There are two main approaches for solving RL problems: methods based on value functions and policy gradients methods. A hybrid of both methods is the actor-critic approach that employs both value functions and policy search. A policy $\pi$ defines the agent's behavior by mapping states to actions. A value function provides an estimation of the future return. This information can then be used to update the policy (Sutton and Barto [58]).

Deep RL uses deep neural networks as function approximator for scaling to previously intractable, high-dimensional problems (e.g. working with images). The first successful deep RL algorithm was deep Q-network (DQN) [5], which achieved human level performance on many Atari video games by learning directly from pixels. Although DQN and its extensions succeeded at solving problems of high-dimensional state spaces (e.g. pixels), they can only handle discrete, low-dimensional action spaces (e.g. left, right). Driving a car requires continuous actions (steering, throttle). The relevant algorithms are discussed below.

**Deep Deterministic Policy Gradient (DDPG).** By combining the insights of DQN with the actor-critic deterministic policy gradient algorithm, DDPG [7] allows for solving a wide variety of continuous control tasks. DDPG utilizes an actor function $\mu(s|\theta^\mu)$, specifying the current policy, and a critic function $Q(s, a|\theta^Q)$, both approximated by neural networks. At each step, based on the current state $s_t$, the agent chooses an action according to $a_t = \mu(s_t|\theta^\mu) + \mathcal{N}$, with a noise process $\mathcal{N}$ to allow for exploration, and obtains a reward $r_t$ and a new state $s_{t+1}$ from the environment. The observed transitions $(s_t, a_t, r_t, s_{t+1})$ are stored in a replay buffer. At each step, a minibatch of $N$ transitions is uniformly sampled from the buffer. The parameters

of the critic network are then optimized using Adam optimization to minimize the loss given as:

$$L(\theta^Q) = \frac{1}{N} \sum_{i=1}^{N} (y_i - Q(s_i, a_i|\theta^Q))^2 \qquad (1)$$

$$y_i = r_i + \gamma Q'(s_{i+1}, \mu'(s_{i+1}|\theta^{\mu'})|\theta^{Q'}) \qquad (2)$$

where $y_i$ is the one-step target with the discount factor $\gamma$. Here, $Q'(s, a|\theta^{Q'})$ and $\mu'(s|\theta^{\mu'})$ are the target networks associated with $Q(s, a|\theta^Q)$ and $\mu(s|\theta^\mu)$. Their parameters are updated at each step using soft updates, i.e. $\theta' \leftarrow \tau\theta + (1 - \tau)\theta'$ with $\tau \ll 1$. To update the parameters of the actor network, a step proportional to the sampled gradient of the critic network with respect to the action is taken, which is given by:

$$\nabla_{\theta^\mu} J \approx \frac{1}{N} \sum_{i=1}^{N} \nabla_a Q(s, a|\theta^Q)|_{s=s_i, a=\mu(s_i)} \nabla_{\theta_\mu} \mu(s|\theta^\mu)|_{s=s_i}. \qquad (3)$$

In summary, at the start of the training, the replay buffer is empty and the weights of the actor and critic are randomly initialized. The agent proceeds to issue actions and the resulting observed state and reward are added to the replay buffer. The model samples the replay buffer to improve the critic function and consequently, the actor function. When the episode finishes, the environment starts a new episode and training continues until convergence.

## Algorithms and training

We trained the autonomous driver using DDPG. We choose the DDPG algorithm since it can successfully address problems in the continuous actions space and has a good sample efficiency (samples needed to convergence). At each time step, the agent receives detailed information about the state of the environment. However, as many parts of the simulation are not directly accessible to the agent, the environment is partially observable even if only a single car is on the track.

Interaction with the environment takes place in discrete time steps with a spacing of 50ms (20hz). The input for the algorithms consists of vehicle telemetry data. Although the simulator (VSM) can provide up to 800 channels of telemetry, we carefully selected the channels that are more relevant to the model and the ones that depend on the car state rather than on the track (e.g., we didn't include the absolute position). Selecting these channels allows the method to be generalized and extended to other tracks. Table 5 shows the channels used.

The output of the model consists of continuous variables that range from zero to one for each control of the car (throttle pedal, brake pedal and steering wheel). The gear shift and the clutch were automated. No DRS (Drag Reduction System) was used. The fuel consumption and the tires' wear were disabled. The car setup and characteristics and the track were the same as the ones in the user study in Part 1: "Quantifying human driving performance". The reward is calculated as:

$$r = V_x(1 - |\text{distance to track axis}|) \qquad (4)$$

This function maximizes the velocity in the direction of the track axis, and penalizes the agent for not following the track axis. The track axis was given by the racing line of the best lap time from the part 1. The racing line is defined as a sequence of points on the track. The car starts in the exit of the last corner and the episode finished after completing a full lap or when the car was out of track.

**Table 5. Telemetry features used as input.**

| Notation | Description |
|---|---|
| $\theta$ | Angle between the car direction and the direction of the track axis or racing line. |
| track | Distance between the car and the track axis or racing line. |
| $V_x$ | Speed of the car along its longitudinal axis. |
| $A_l at$ | Acceleration in the lateral axis from the car point of view |
| $A_l ong$ | Acceleration in the longitudinal axis from the car point of view |
| *Floatangle* | Slip angle of the body |
| *Yawrate* | Angular motion in the vertical axis |
| *TyreSaturationLateralFL* | Saturation of the front left tire. 100% represent full saturation |
| *TyreSaturationLateralFR* | Saturation of the front right tire. 100% represent full saturation |
| $f_{rot}$ | Number of rotations per minute of the car engine. |
| *Actualcurvature* | Curvature of the car |
| *Demandcurvature* | Curvature of the track at the current car position |
| LAC | Look ahead curvature. Vector of 4 curvature measurements from the racing line at 20, 40, 60 and 80 meters ahead. |

Description of the telemetry features used as input for the RL model.

The level of detail of the physics engine comes with the cost that each step of the simulation ends up being rather computational expensive. An offline computation of a lap takes about a fifth of running the same lap in real time. This means that, for example, the time to make a lap in Barcelona circuit while driving in real time is about 100s, while computing the same lap off-line takes around 20s.

Our first resulting models had oscillations in the three controls (steer, throttle and brake). The oscillations were reduced by including to the state two past actions taken by the model. Adding previous actions to the state gave the model a notion of previous actions.

We used the same hyper-parameter settings as in the original DDPG paper [7]: Adam optimization was used with a learning rate of $10^{-4}$ and $10^{-3}$ to learn the actor and critic networks respectively. Networks were trained with a batch size of 256 and a discount factor of $\gamma = 0.99$. For the soft target updates we used $\tau = 0.001$. The actor network had two hidden layers. Compared to the original DDPG, our critic network had a more complex structure. Actions were processed through one hidden layer and the states through two hidden layers. The sum of both streams was processed by another hidden layer. Furthermore, a replay buffer was used with a size of $10^6$, since we found this leads to a more stable and higher training performance in terms of episode reward. We also used the end of episode correction as reported in [3].

## Model selection

Our autonomous driver evolves over time while trying to optimise for the target reward function. So, that means that overtime the AI will go through different performance levels. We propose to extract laps at each performance level defined in part I, based only on lap-time and then compare features to see if the level of performance of the AI can also be characterised by the features we had for humans in the same way. We extract the following features for each group: the simulation steps needed to perform as the slowest lap (entry) in the group, the number of laps needed to have a similar performance as the slowest lap, fastest and slowest lap times for the group, lap measures (performance, vehicle control) for the autonomous driver. The groups were taken from the clustering of part I as shown in Table 2.

**Table 6. Model performance across the different groups.**

| Group | Best group LT | Worst group LT | #Laps | Steps | Wall Time | Group enter LT |
|---|---|---|---|---|---|---|
| Worst | 168.442 | 211.266 | 2315 | 208473 | 1.737 | 158.651 |
| Bad | 145.716 | 168.442 | 2315 | 208473 | 1.737 | 158.651 |
| Medium | 130.372 | 145.716 | 2353 | 254515 | 2.121 | 139.150 |
| Good | 118.825 | 130.372 | 2374 | 284074 | 2.367 | 129.101 |
| Very good | 107.680 | 118.825 | 2462 | 480655 | 4.005 | 117.701 |
| Excellent | 99.080 | 107.680 | 3347 | 2962279 | 24.686 | 107.100 |
| %50 Excellent | 99.080 | 101.680 | 9001 | 8449216 | 70.410 | 101.600 |
| %25 Excellent | 99.080 | 99.800 | 10750 | 12819312 | 106.828 | 99.749 |
| Best Excellent | 99.080 | 99.080 | 14153 | 16004610 | 133.254 | 98.900 |

Lap time of the RL model: Best Excellent is the best lap time of the excellent group; %25 Excellent is the lower boundary of the %25 of laps from the excellent group; %50 Excellent means the lower boundary of the %50 of the laps from the excellent group; Best and worst group lap time: best and worst human lap of the group. #Laps, Steps and Wall Time: the number of laps, the number of simulator steps and the wall time that the model needed to enter the group. Group enter LT: Lap time when the model entered the group.

## Results

Table 6 shows the best lap times of the autonomous driver and the humans drivers in each group. It also shows the progress of the algorithm while training. The best lap time achieved by the model was 98.91 s vs 99.08 s of the best human lap. This is 0.18 s faster than the best human lap. The RL agent reached the *EXCELLENT* group after twenty hours of training and was ranked first in the total raking of laps after three days. We can calculate an ideal lap as the sum of each best sector times. In the case of the autonomous driver the ideal lap time was 98.40 s (vs. the human ideal lap is 97.7 s). Fig 14 shows the progress of the algorithm while training. The colors represent model lap that falls into the different groups. The discontinuity from laps 4000 to 8000 are due to instabilities in the DDPG algorithm. Note that there is a comparably large number of steps to move from level *VERYGOOD* to *EXCELLENT*, larger than all the steps needed to reach level 5. Fig 15 shows the learning progress at each sector.

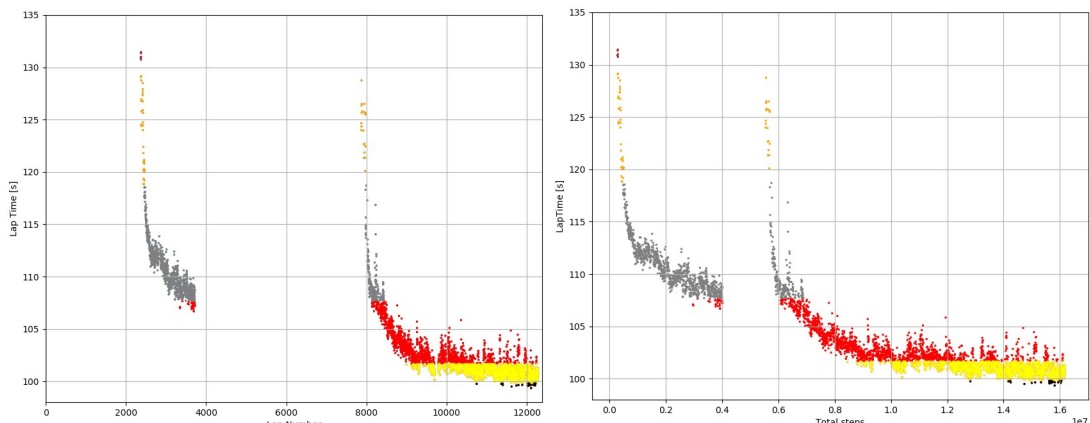

**Fig 14. Progress the algorithm while training.** The y-axis is the lap time and the x-axis is the total steps seen by the model. Brown dots are laps that fall in the "bad" lap group form Part I. Orange are the middle group. Gray are the good laps and red the excellent. Yellow and black are the best 50% and 25% of the excellent group, respectively. Right: the x-axis represents the number of simulated steps. Left: The x-axis represents the number of laps.

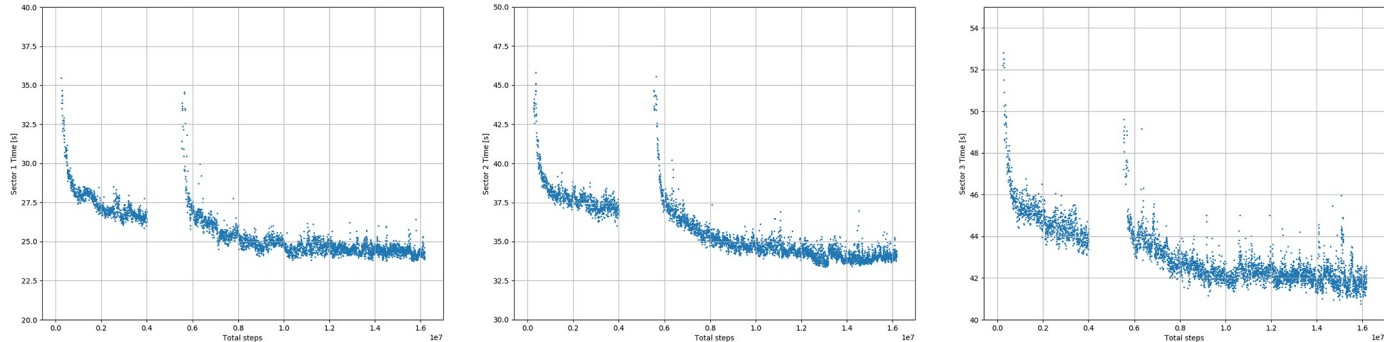

**Fig 15. Progress of the algorithm while training in each sector of the track.** Sectors I, II, III correspond to the right middle and left images. The x-axis represents the number of laps and the y-axis represents the time to complete the Sector.

Sector I is very consistent while sector III has a higher variance as expected since it is the most challenging sector in the track.

## Comparison between best algorithm lap and best human lap

The algorithm performed a lap time of 98.91 s whereas the best human driver lap time was 99.08 s (+0.18 s). Fig 16 shows the telemetry of the best lap of the algorithm and the best lap of the human drivers in sector I of Barcelona circuit. The algorithm sector time was 22.92 s and the human one was 23.42 s. The saturation channel shows the percentage of tire utilization, when 100% is reached the tire will start slipping. This means that keeping the saturation to 100% represents the ideal utilization of the tires capabilities. The AI driver keeps the tire saturation to 100% most of the time while cornering, while the human drivers fully utilized the tires only in some parts of the corner and having to use more time to brake than the AI driver. The brake channels shows that the human driver brakes very smoothly while the AI driver

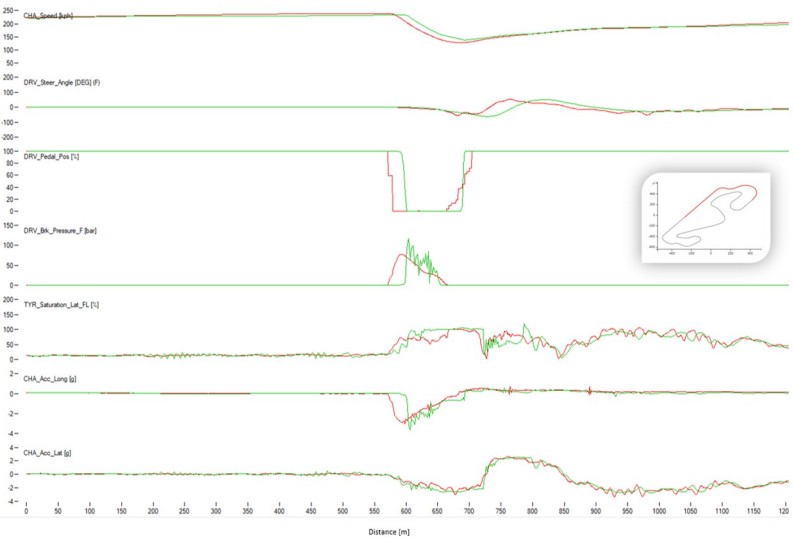

**Fig 16. Comparison between the best human lap (red) and the best result of the algorithm (green).** The image shows the Sector I of Barcelona. The X-axis represents distance through the racing line. The Y-axis (from top to bottom): Vehicle speed, steering wheel angle, throttle position, brake position, front left tire saturation, lateral acceleration, longitudinal acceleration.

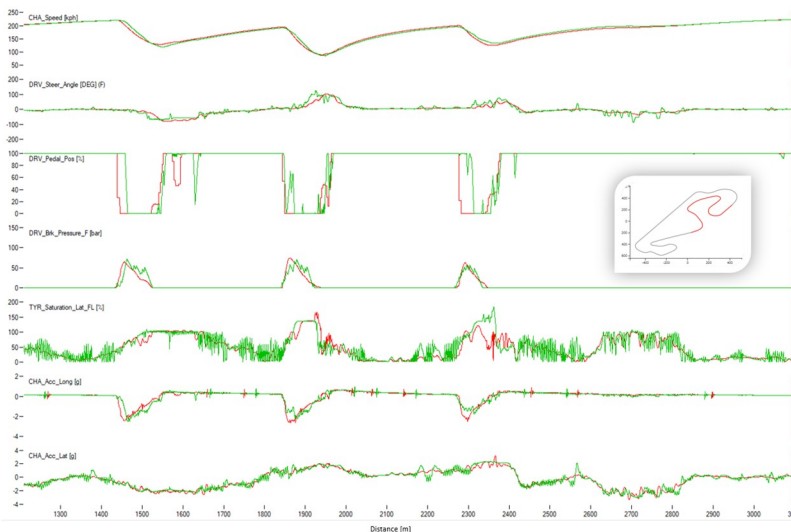

**Fig 17. Comparison between the best human lap (red) and the best result of the algorithm (green).** The image shows Sector II of Barcelona. The X-axis represents distance through the racing line. The Y-axis (from top to bottom): Vehicle speed, steering wheel angle, throttle position, brake position, front left tire saturation, lateral acceleration, longitudinal acceleration.

modulates the brakes in order to keep the saturation of the tires at 100%. The throttle position is very abrupt and sharp for the AI driver, and the human drivers' transition to full throttle is smoother. Figs 17 and 18 show Sector II and corner 10 of Barcelona circuit, respectively. In general, the pattern is similar to that in Sector I. The A.I. tends to brake later and uses the tires better.

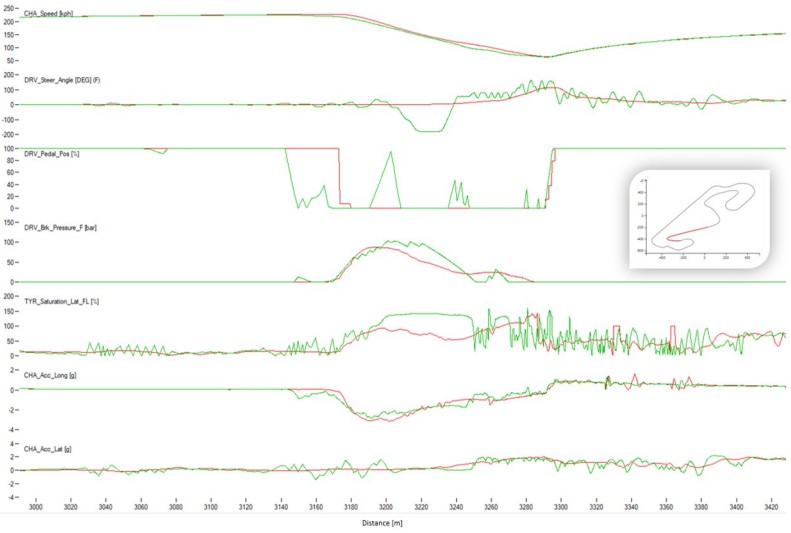

**Fig 18. Comparison between the best human lap (red) and the best result of the algorithm (green).** The image shows the corner 10 of Barcelona. The X-axis represents distance through the racing line. The Y-axis (from top to bottom): Vehicle speed, steering wheel angle, throttle position, brake position, front left tire saturation, lateral acceleration, longitudinal acceleration.

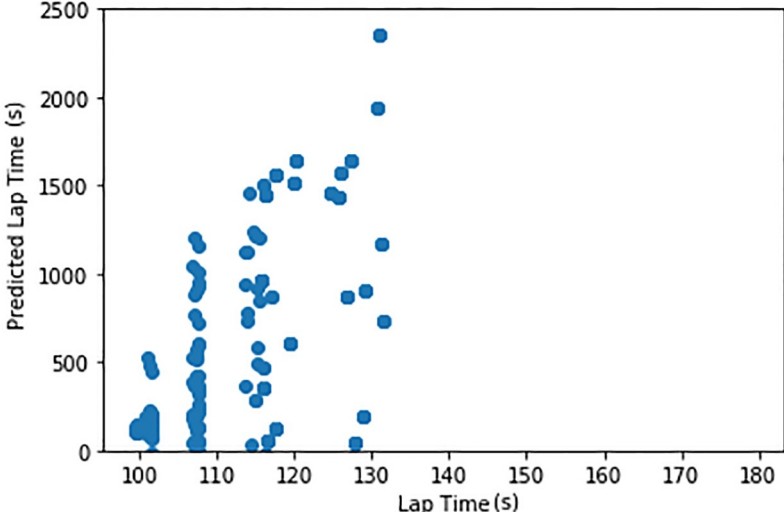

**Fig 19. Linear regression predicting the lap time of the RL agent laps by training the model trained on the laps of human drivers.**

## 0.1 Comparing driving behaviors of humans and RL agents

We attempted to establish whether an RL agent performing a lap follows similar patterns to human drivers. To that end, we computed the same features as described above, trained a linear regression with the data from human laps and tested how well the model can predict laps of the RL agent. Theoretically, if in both cases the same driving behaviors are utilized, a model trained on human driver laps should be able to predict the laps performed by RL agents.

The results are presented in Fig 19, which demonstrates that the model fails to predict any meaningful lap time, suggesting that the RL agent follows entirely different patterns when learning to drive.

To confirm this assumption, we explored how well the lap time of laps performed by the RL agent can be predicted. For this purpose, we split the RL agent laps in train (70%) and test set (30%). We trained a linear regression model using the train set of RL laps and tested on the test set. The same features as in the case of human laps are used. The prediction results presented in Fig 20 demonstrate that the linear regression model predicts the lap time with an absolute accuracy of 99.4% or an absolute error of 0.62 seconds (0.06%), which is more accurate than the model predicting the lap time of human drivers.

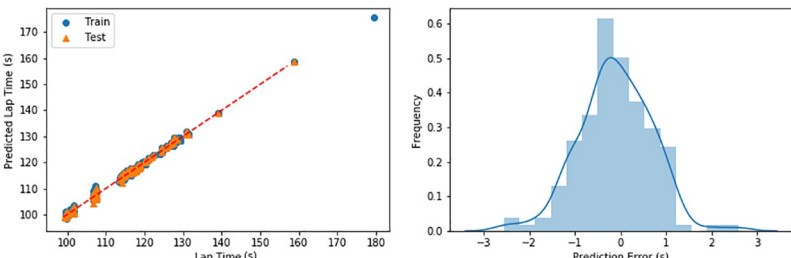

**Fig 20. Linear regression predicting the lap time of RL agents based on the selected features (left) and the histogram of prediction error (right) on the test set.**

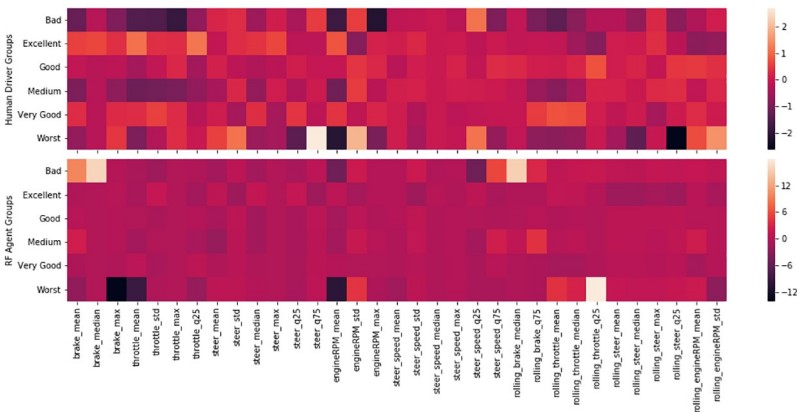

**Fig 21. Scaled features of the laps performed by human drivers and RL agent.** The laps are grouped based on the performance levels. The features are scaled to have a mean set to zero and a standard deviation set to one.

While the model trained on laps performed by humans predicts the lap time of RL agent laps very poorly, models trained on either human laps or RL agent laps can predict the lap-time of their own kind quite accurately. This finding suggests that driving behavior patterns of humans and RL agents are quite different. That could also be observed based on the weights of each linear regression model (see Table 4), which reveal large variations among the models trained on humans and RL agent laps. The same conclusion could be drawn from the heat map of the features presented in Fig 21, which shows that the extracted features differ between human and agent laps. In the case of RL, features describing the braking behavior take the first four places in terms of relevance to the model, which reflects the complexity of learning the correct braking behavior.

## Lessons learned

This work investigates driving patterns of human drivers and self-driving agents in a racing environment. The ultimate goal of this research line is cyclic: first, patterns and behaviors gathered from human drivers are used to improve the performance of self-driving agents; and, second, the results for high-performing self-driving agents are employed to help humans perform their job more efficiently and successfully (e.g., drivers to race, trainers to train the drivers, manufactures to evaluate the configurations and the various parameters of the car). While our ultimate research goal very broad and ambitious and will require multiple studies, the work presented in this paper makes important steps towards achieving it.

Our engineered features were able to predict the success of a race quite accurately, indicating that they capture the crucial elements of a race that determine the racing quality of driving. Thus, they should be considered and assessed when developing self-racing agents.

Moreover, our work showed that driving patterns of human drivers and self-driving agents are very different and that different features are more important in each category of drivers. Therefore, there is room for improving self-driving agents and enforcing some degree of adopting the human drivers' behaviors. We aim to develop the relevant methods and further investigate this in our future work. Such approaches would be very valuable in some use cases, e.g., when the purpose of a self-driving agent is to discover new driving strategies to be adopted by a human driver. Under this scenario, it would be essential that actions of the self-driving agent are feasible for the human driver as well. As our analysis showed, the self-driving agent can manipulate the controls (steering, pedal, brake) much faster than a human one. Humans

have a limited capacity of adopting such patterns due to their motor skills and have a limited reaction time, which puts them in disadvantage. Therefore, any strategy that originates from agents and requires manipulating controls in a way that it is physically unattainable for humans is not useful for training purposes. The developers of self-driving agents that target such use cases should consider these differences and possibly introduce some artificial restrictions to the self-driving agent to be able to apply the self-driving output to human driving. In other use cases, e.g., when the self-driving agent is trying to win a racing contest against other self-driving agents, such restrictions would not be necessary.

Last but not least, our results demonstrated that driving behaviors in some segments were much more important to the success of the entire lap than others. Thus, self-driving agents could potentially improve their overall performance if they were more extensively trained on the most important segments.

## Conclusion

In this work, we carried out an extensive analysis of telemetry data from laps completed by human drivers. Examining the performance of human drivers during the laps and the associated NASA-TLX, it was found that participants in the *EXCELLENT* performance level reported lower perceived effort than those in the *VERY GOOD* and *GOOD* levels. Our assumption is that the driver makes a mental plan where the car has to be in order to achieve the best possible performance, but as the real execution differs the driver is trying to adjust while perceiving the errors. A driver performing at the best level perceives success in the formulated plan, which is translated in less correction effort. However, with the current evidence, this is merely an assumption and more in-depth study is needed to investigate cognitive effects. Connecting the NASA-TLX Effort result with the RL algorithm performance, Fig 14 shows the algorithm effort in terms of steps and laps at each performance level. In these terms, 1/3 of the training is spent until the agent can drive the track at *GOOD* level, but it takes the algorithm 2/3 of the training time to progress from good to excellent. Thus, the comparable effort increases to progress at higher performance levels.

Based on the data obtained from laps driven by humans, we analyzed telemetry information and estimated which feature has more impact on the driver's performance. This outcome lead to the creation of a model for predicting the final performance during the lap in terms of lap-time before finishing the lap. We trained the model on human and RL agent laps and achieved an accuracy of about 97% and 99%, respectively. Models trained on either human laps or RL agent laps were able to predict the lap-time of their own kind quite accurately, suggesting that driving behavior patterns are rather different between humans and RL agents.

Since our lap-time and corner prediction of the RL agent lap was quite precise, one avenue for future work is to use it as a reward function in a sparse rewards RL setting or as the cost function of MPC (model predictive control) algorithms. Moreover, we demonstrated the impact that each feature has to the predicted lap time. This information can be applied to formulate new reward functions that take the features into account, so as to guide the model towards the desired behavior. When analyzing the features during the AI lap, we established that the brake affects the lap quality most. This indicates that the AI needs many more laps to learn how to brake than it needs to learn how to steer and accelerate. This behavior was also observed in humans. An interesting outcome of this analysis is that the model can help to identify channels that are more difficult to master. This knowledge can be used to create maneuvers to train humans or AIs more efficiently.

The differences that this study reveals could potentially lead to improving the performance of both the autonomous and the human drivers. The RL driver in our case took over from a

human by using the racing line of the best human lap as the reward and worked tirelessly until a fastest lap time was achieved. This could be the basis for an improvement strategy for human drivers, for example, by suggesting a more aggressive brake throttle relation in Sector 1, as as shown in our study.

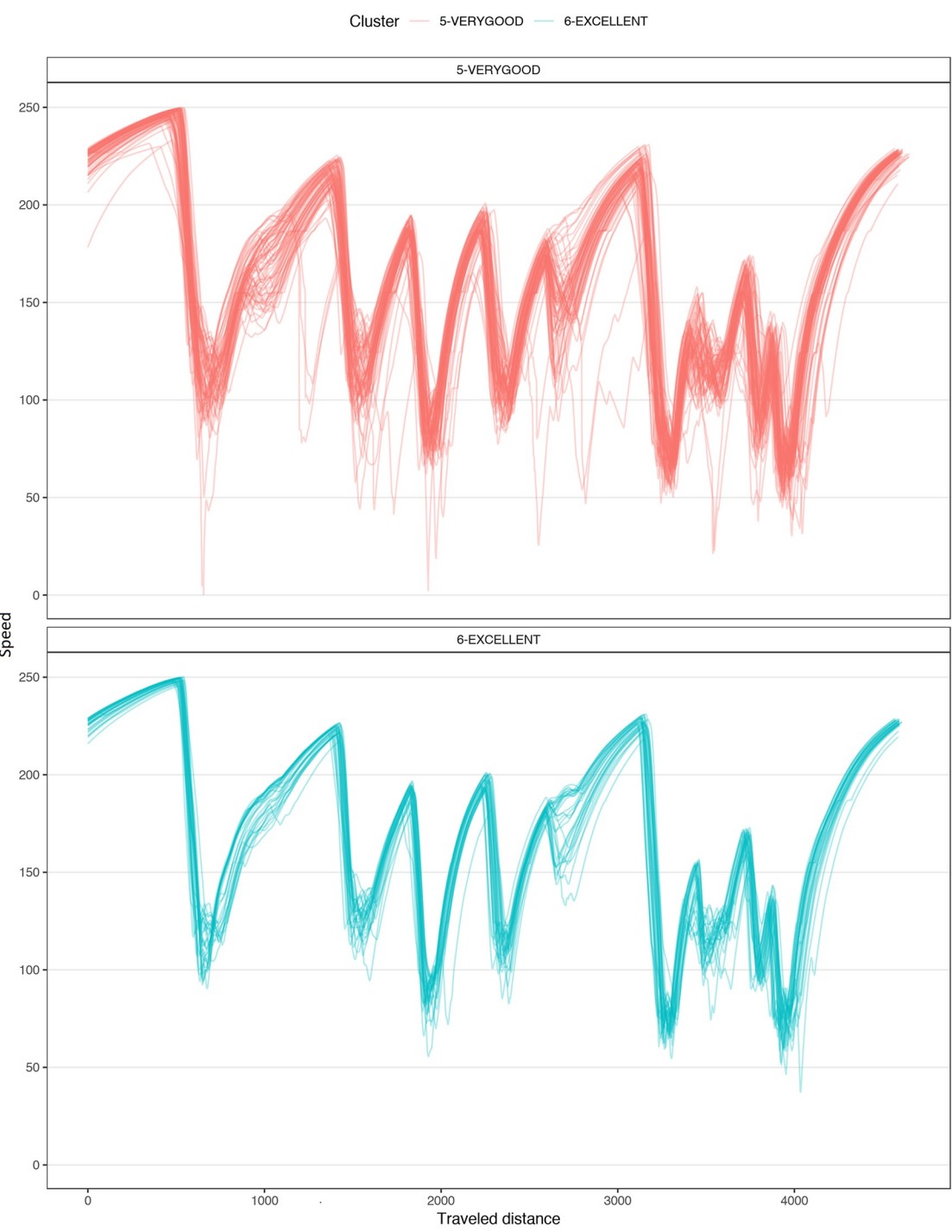

**Fig 22. Speed channel comparison between all the laps from group EXCELLENT and all the laps from group *VERYGOOD*.**

Furthermore, we demonstrated that RL can deal with the task of driving a professional simulated racing vehicle entirely from scratch merely by giving a notion of long-term reward. To establish a solid baseline, we validated the performance via an extensive comparison with human drivers in order to have a solid baseline. Our model joined the "excellent" group after

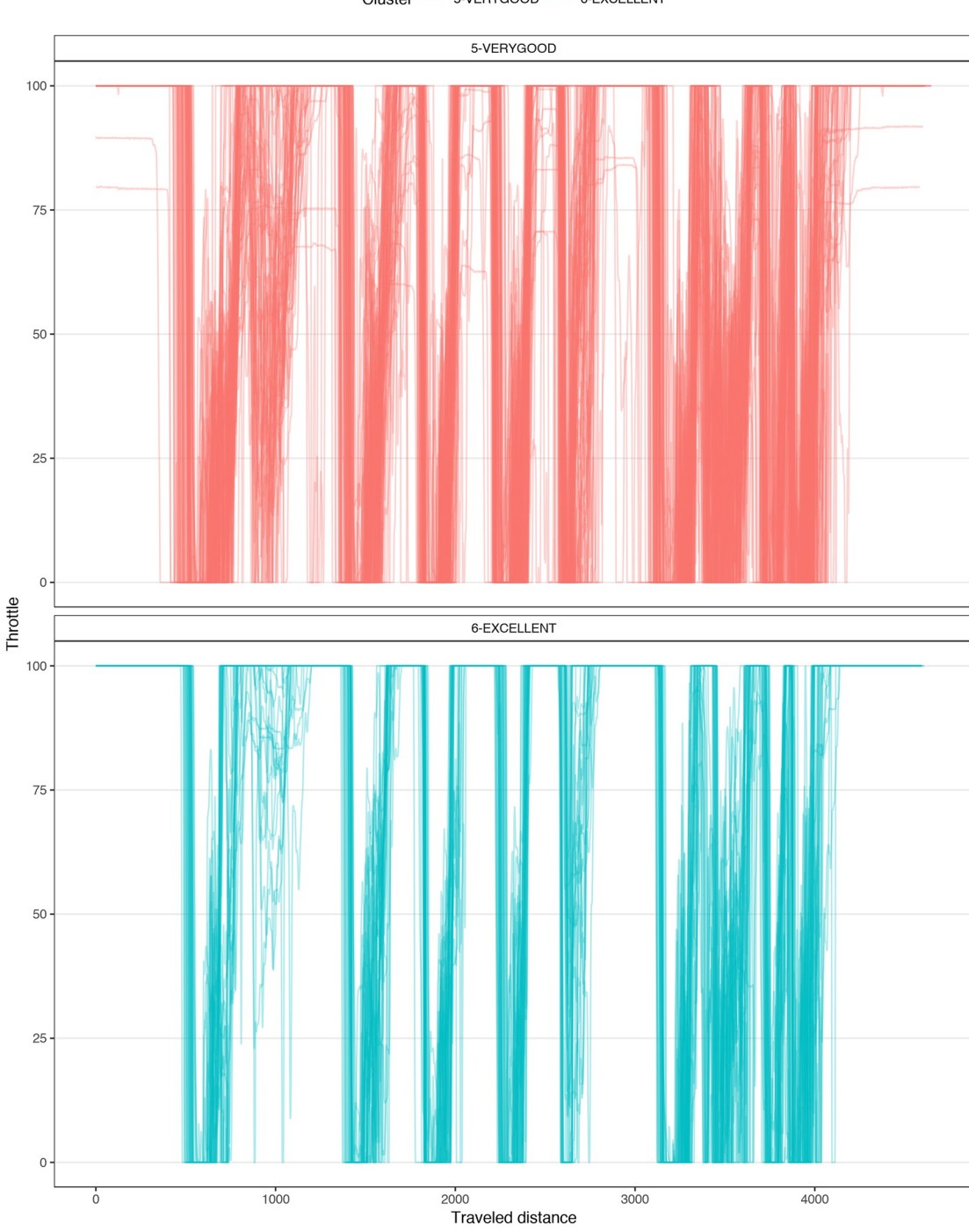

**Fig 23. Throttle position comparison between all the laps from group EXCELLENT and all the laps from group *VERYGOOD*.**

twenty hours of training and achieved super-human performance after three days. Although the driver model yielded excellent results, there is still room for improvement towards the ideal performance. End-to-end RL is not sufficient for driving the ideal lap on a professional simulator. A better representation of the current and future states seems to be necessary. Observing the ideal lap times put together with the fastest times per sector indicates that

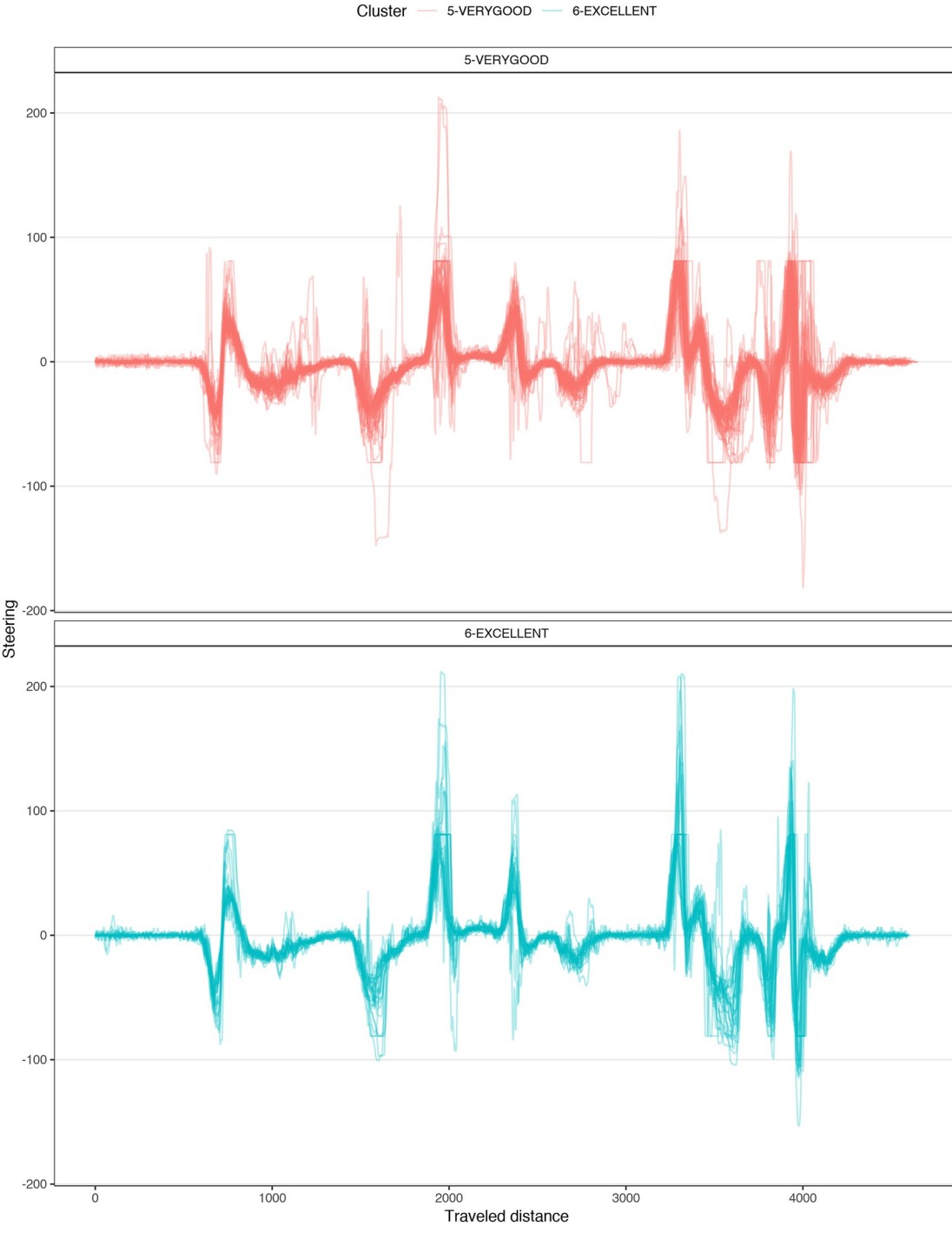

**Fig 24. Steering wheel comparison between all the laps from group EXCELLENT and all the laps from group *VERYGOOD*.**

training the sectors separately may lead to better results. Curriculum learning and inverse RL may be able address this issue. Our work opens future research directions, including creating models for improving the reward function which will allow us to improve the sample efficiency and final performance of the autonomous driver.

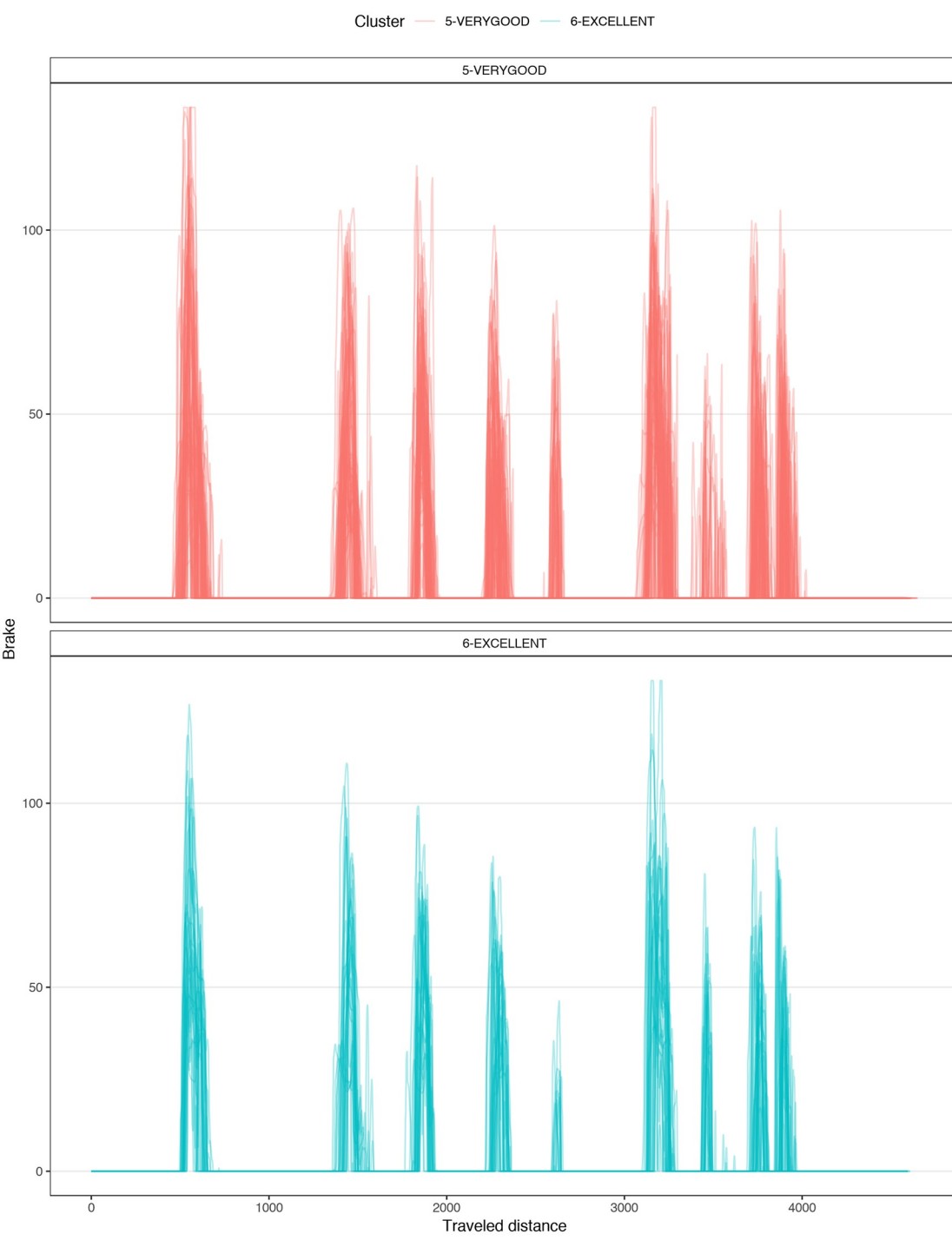

**Fig 25. Brake comparison between all the laps from group EXCELLENT and all the laps from group *VERYGOOD*.**

## 1 Future work

Although DDPG has a good sample efficiency (samples needed for convergence), it has the disadvantage that it can use only one CPU core at the time when collecting simulator experience. We used DDPG for simplicity reasons in this study and we plan to use newer algorithms

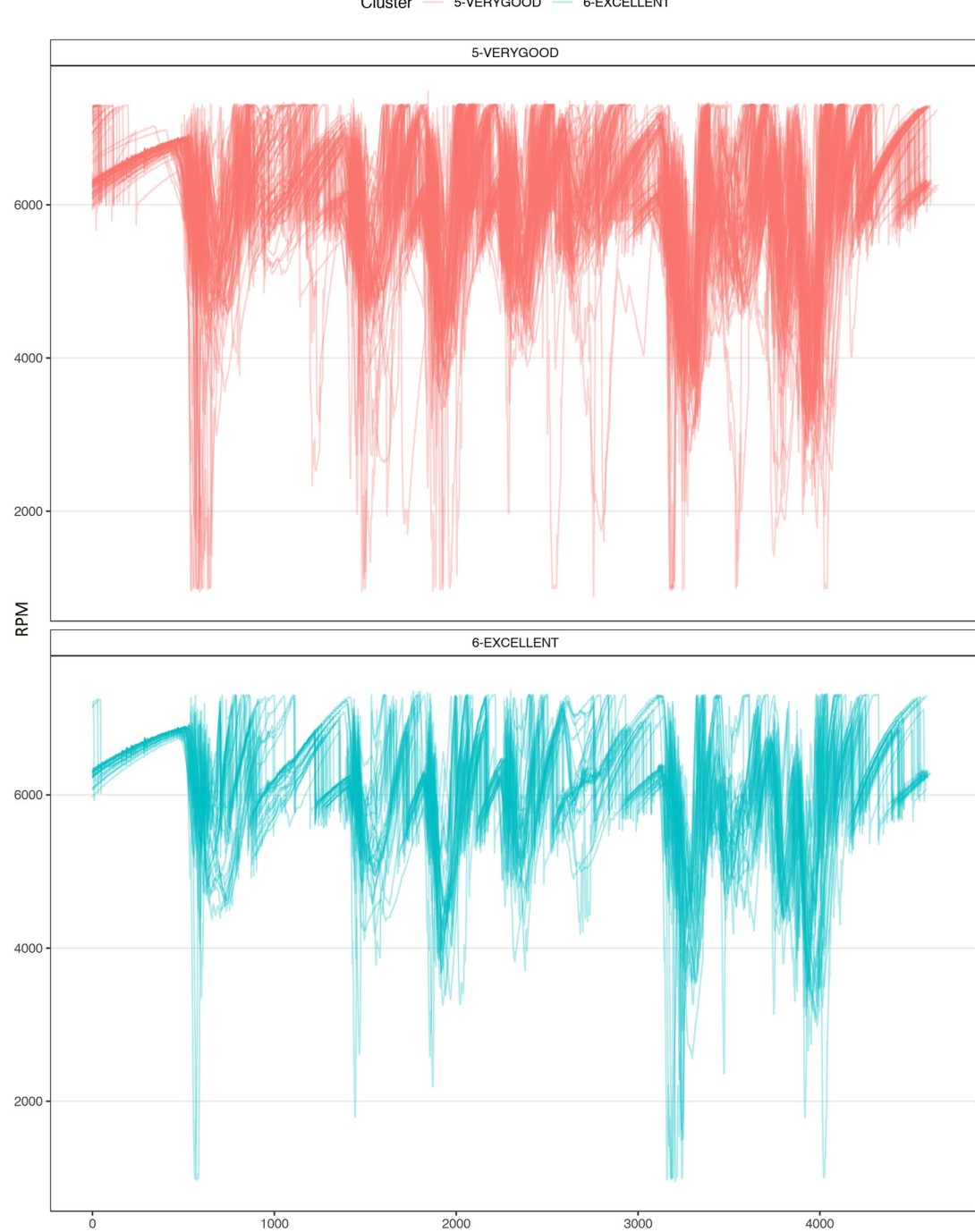

**Fig 26. Engine RPMs comparison between all the laps from group EXCELLENT and all the laps from group *VERYGOOD*.**

that exploit parallel computing, such as A3C and PPO (Proximal Policy Optimization) [59, 60].

## Appendix

### Vehicle control features

The Figs 22–26 show a comparison between all laps from group *EXCELLENT* and all laps from the group *VERYGOOD* of the channels speed, throttle, steering wheel, brake and engine RPMs. We observe that the laps from the group EXCELLENT are more consistent than the driver in the group *VERYGOOD* in the speed and throttle channel. That means that the drivers of the group *EXCELLENT* tend to repeat their speed profile and their sequence of throttle command closely every lap. This is also true even for different drivers.

There is not much variability across the laps in the brake channel. The RPMs are highly variable even on their own group.

## Supporting information

**S1 File.**
(PDF)

**S2 File.**
(PDF)

## Author Contributions

**Conceptualization:** Adrian Remonda, Granit Luzhnica.

**Data curation:** Adrian Remonda, Eduardo Veas.

**Formal analysis:** Adrian Remonda, Eduardo Veas, Granit Luzhnica.

**Funding acquisition:** Eduardo Veas.

**Methodology:** Adrian Remonda.

**Resources:** Eduardo Veas.

**Software:** Adrian Remonda.

**Writing – original draft:** Adrian Remonda.

**Writing – review & editing:** Adrian Remonda, Eduardo Veas, Granit Luzhnica.

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
