## [Decision Letter · Decision Letter 0]

19 May 2020

PONE-D-20-06965

Comparing Driving Behavior of Humans and Autonomous Driving in a Professional Racing Simulator

PLOS ONE

Dear Mr Remonda,

Thank you for submitting your manuscript to PLOS ONE. After careful consideration, we feel that it has merit but does not fully meet PLOS ONE’s publication criteria as it currently stands. Therefore, we invite you to submit a revised version of the manuscript that addresses the points raised during the review process.

We would appreciate receiving your revised manuscript. To enhance the reproducibility of your results, we recommend that if applicable you deposit your laboratory protocols in protocols.io, where a protocol can be assigned its own identifier (DOI) such that it can be cited independently in the future. For instructions see: http://journals.plos.org/plosone/s/submission-guidelines#loc-laboratory-protocols

We look forward to receiving your revised manuscript.

Kind regards,

Chen Lv, PhD

Academic Editor

PLOS ONE

Journal Requirements:

3. Thank you for stating the following in the Competing Interests/Financial Disclosure section* (delete as necessary):

"The funding for this work was awarded to Adrian Remonda. This research was partially

funded by AVL GmbH (www.avl.com) and Know-Center GmbH (www.know-center.at).

Know-Center is funded within the Austrian COMET Program - Competence Centers for

Excellent Technologies - under the auspices of the Austrian Federal Ministry of

Transport, Innovation and Technology, the Austrian Federal Ministry of Economy,

Family and Youth and by the State of Styria. COMET is managed by the Austrian

Research Promotion Agency FFG (https://www.ffg.at/en). The sponsors and funders

played no role in the study design, data collection and analysis, decision to publish, or

preparation of the manuscript."

We note that one or more of the authors have an affiliation to the commercial funders of this research study : "Know-Center GmbH"

b) Please also provide an updated Competing Interests Statement declaring this commercial affiliation and your additional commercial funder (AVL GmbH)  along with any other relevant declarations relating to employment, consultancy, patents, products in development, or marketed products, etc. 

Additional Editor Comments (if provided):

Based on reviewers' comments, a major revision is required. Please carefully address the concerns from both reviewers.

Reviewers' comments:

Reviewer's Responses to Questions

**Comments to the Author**

1. Is the manuscript technically sound, and do the data support the conclusions?

Reviewer #1: Yes

Reviewer #2: Yes

2. Has the statistical analysis been performed appropriately and rigorously? 

Reviewer #1: Yes

Reviewer #2: Yes

3. Have the authors made all data underlying the findings in their manuscript fully available?

Reviewer #1: Yes

Reviewer #2: Yes

4. Is the manuscript presented in an intelligible fashion and written in standard English?

Reviewer #1: Yes

Reviewer #2: Yes

5. Review Comments to the Author

Reviewer #1: This paper compares the behavior of autonomous and human drivers in aggressive driving scenario based on a professional racing simulator. 14 participants were recruited and grouped to drive to achieve the fastest lap, keeping the car inside the track according to performance (lap-time), defining driving behaviors at different performance levels. An extensive analysis was performed on vehicle control features obtained from telemetry data with the goal to predict driving performance aiming to inform an autonomous system. A state-of-the-art reinforcement learning algorithm was trained to control the brake, throttle, and steering of the simulated racing car.

In general, this is an interesting research study. However, I have several concerns and some comments for authors to consider to improve the quality of this paper:

1. Authors need justify the sample size selection, i.e., 14 participants.

2. What's the motivation for autonomous driving for race cars?

3. Authors have briefly reviewed the state of the art on race cars. But I do suggest to add one section to review the general state of the art in regular car driving simulator research and this will be beneficial to researchers.

4. The autonomous driving part is difficult to follow. Authors may be reorganize that session to ensure it is easy to follow.

5. In all figures I suggest to add units for horizontal and vertical axes.

6. Was the autonomous driving, reinforcement machine learning session, trained by the results of human drivers (participants)? If so what's the meaning of comparing their driving behaviors?

Reviewer #2: The author proposed an analysis study for the comparison of human driving and machine driving performance for Motorsports. The study is well written and interesting, however, I think the author can further clarify the work from the following aspects.

1. As far as I see, the authors do not completely answer the question they proposed in the contribution part, or make readers confused. For example, the question like "what can we learn from humans to boost the machine performance" need to further discussed as currently only limited conclusion can be made.

2. The literature review part show the authors gain a good knowledge in the motor sports and racing, and the RL algorithms, etc. However, one important part is missing, which are review of existing studies on the analysis between human and machine, particular some important conclusions, connections, and methodologies.

3. More discussions and data visualization are expected for the comparison of the human performance and machine performance. For example, for the Barcelona-Catalunya circuit, is there any moment that human and automation show similar driving performance or characteristics, and when will the performance be significant different with each other?

4. The authors are encourage to summarize the most important knowledge that human driving can be used to improve the machine training, and discuss any application limitation.

5. The manuscript need to be carefully checked in terms of English writing and citation. For example, in page 20, line 618, which table is refer to?

6. PLOS authors have the option to publish the peer review history of their article (what does this mean?). If published, this will include your full peer review and any attached files.

Reviewer #1: Yes: James Yang

Reviewer #2: No

---

## [Author Response · Author response to Decision Letter 0]

8 Oct 2020

We appreciate the comments of the reviewers. We have uploaded the response as a pdf with the name "Response to Reviewers"

---

## [Decision Letter · Decision Letter 1]

26 Oct 2020

PONE-D-20-06965R1

Comparing Driving Behavior of Humans and Autonomous Driving in a Professional Racing Simulator

PLOS ONE

Dear Dr. Remonda,

Thank you for submitting your manuscript to PLOS ONE. After careful consideration, we feel that it has merit but does not fully meet PLOS ONE’s publication criteria as it currently stands. Therefore, we invite you to submit a revised version of the manuscript that addresses the points raised during the review process.

We look forward to receiving your revised manuscript.

Kind regards,

Chen Lv, PhD

Academic Editor

PLOS ONE

Additional Editor Comments (if provided):

Please carefully address reviewer's further concerns.

Reviewers' comments:

Reviewer's Responses to Questions

**Comments to the Author**

1. If the authors have adequately addressed your comments raised in a previous round of review and you feel that this manuscript is now acceptable for publication, you may indicate that here to bypass the “Comments to the Author” section, enter your conflict of interest statement in the “Confidential to Editor” section, and submit your "Accept" recommendation.

Reviewer #1: (No Response)

2. Is the manuscript technically sound, and do the data support the conclusions?

Reviewer #1: Partly

3. Has the statistical analysis been performed appropriately and rigorously? 

Reviewer #1: No

4. Have the authors made all data underlying the findings in their manuscript fully available?

Reviewer #1: No

5. Is the manuscript presented in an intelligible fashion and written in standard English?

Reviewer #1: Yes

6. Review Comments to the Author

Reviewer #1: I have more comments for authors to address as follows:

1. In line (18) and line (35), the authors tried to discuss about passenger self-driving cars as they are less risky to magnify that the racing cars are comparatively in their maximum risk ceiling. But the fact that self-driving cars still operate in numerous stochastic environmental conditions including perception of the surrounding (object detection, identification, pedestrian and road cue detection etc), multiple road structures and routes, complicated mix of vehicles and interaction with other human-driven vehicle might makes them as complex and requires the consideration of numerous parameters. But during racing even though the speed is very high to achieve less lap time, the track does not have no passenger involved and the routes are pre-determined (Figure 1-Page 9/39).

2. Lines (48-65) and (66-83) are repeated.

3. What are the requirements to select TORCS as the main software candidate for simulation as it has many limitations with its Physics engine? (line 317)

4. Participants (444): Based on the result of the simulation, the skilled (2 participants-455) achieve the EXCELLENT time lap as compared to the other 11 participants in a clearer manner. But during clustering (line-481) skill-based behavior was not selected as one to train the RL (Reinforcement learning). Why?

5. Lines (488-493) and lines (482-487) is repeated.

6. In the conclusion and in the body of the article (line 877-879 and lines 534-536), VERY GOOD performance driver’s frustration even is higher as compared to MEDIUM and GOOD performance driver’s from interview and was cross checked with TLX Mental demand. Not clear!

7. In ensuring balanced grouping, the method employed (Kruskal wallis test) was unclear! And what is the result of computing TLX mental demand correlation’s (Figure 3) effect even if the telemetry data of each participants does not match best with RL model?

8. Line (574), breaking should be BRAKING

9. Line (745), we have two ‘the’

10. Even though the Selected model learns to achieve the best time (98.91s -line 778), it still fails to predict any meaningful lap time (line 813 and line 825). If that is so, what is the strong reason to choose RL algorithm to choose to drive the autonomous driver? (850-853)

7. PLOS authors have the option to publish the peer review history of their article (what does this mean?). If published, this will include your full peer review and any attached files.

Reviewer #1: **Yes: **James Yang

---

## [Author Response · Author response to Decision Letter 1]

11 Dec 2020

To thank the reviewers for their fair reviews and detailed list of constructive suggestions. We uploaded an updated the manuscript and answer the reviewer questions in the "Response to Reviewers" document.

---

## [Decision Letter · Decision Letter 2]

29 Dec 2020

Comparing Driving Behavior of Humans and Autonomous Driving in a Professional Racing Simulator

PONE-D-20-06965R2

Dear Dr. Remonda,

We’re pleased to inform you that your manuscript has been judged scientifically suitable for publication and will be formally accepted for publication once it meets all outstanding technical requirements.

Kind regards,

Chen Lv, PhD

Academic Editor

PLOS ONE

Additional Editor Comments (optional):

Reviewers' comments:

Reviewer's Responses to Questions

**Comments to the Author**

1. If the authors have adequately addressed your comments raised in a previous round of review and you feel that this manuscript is now acceptable for publication, you may indicate that here to bypass the “Comments to the Author” section, enter your conflict of interest statement in the “Confidential to Editor” section, and submit your "Accept" recommendation.

Reviewer #1: All comments have been addressed

2. Is the manuscript technically sound, and do the data support the conclusions?

Reviewer #1: Yes

3. Has the statistical analysis been performed appropriately and rigorously? 

Reviewer #1: Yes

4. Have the authors made all data underlying the findings in their manuscript fully available?

Reviewer #1: Yes

5. Is the manuscript presented in an intelligible fashion and written in standard English?

Reviewer #1: Yes

6. Review Comments to the Author

Reviewer #1: Authors have addressed all of my comments and I think it is now acceptable for publication. They have done a good job.

7. PLOS authors have the option to publish the peer review history of their article (what does this mean?). If published, this will include your full peer review and any attached files.

Reviewer #1: **Yes: **James Yang

---

## [Editor Report · Acceptance letter]

11 Jan 2021

PONE-D-20-06965R2 

Comparing Driving Behavior of Humans and Autonomous Driving in a Professional Racing Simulator 

Dear Dr. Remonda:

I'm pleased to inform you that your manuscript has been deemed suitable for publication in PLOS ONE. Congratulations! Your manuscript is now with our production department. 

Kind regards, 

on behalf of

Dr. Chen Lv 

Academic Editor

PLOS ONE